# HiCoM: Hierarchical Coherent Motion for Streamable Dynamic Scene with 3D Gaussian Splatting

**Qiankun Gao[1,2], Jiarui Meng[1], Chengxiang Wen[1], Jie Chen[1,2 ✉], Jian Zhang[1,3 ✉]**

[1]School of Electronic and Computer Engineering, Peking University, Shenzhen, China
[2]Pengcheng Laboratory, Shenzhen, China
[3]Guangdong Provincial Key Laboratory of Ultra High Definition Immersive Media Technology,
Shenzhen Graduate School, Peking University, Shenzhen, China

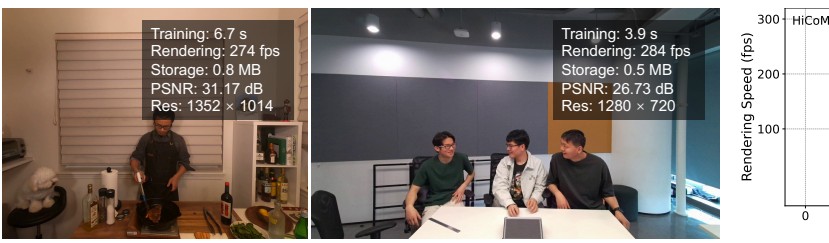

Figure 1: The proposed HiCoM framework for streamable dynamic scene reconstruction achieves competitive rendering quality with significantly shorter training time, faster rendering speed, and substantially reduced storage and transmission requirements. The left figures show results of our HiCoM on N3DV [1] and Meet Room [2] datasets, where "Res" indicates video resolution. The right figure is tested on the N3DV [1] dataset, where the radius of the circle corresponds to the average storage per frame and the method in the top left corner demonstrates the best performance.

## Abstract

The online reconstruction of dynamic scenes from multi-view streaming videos faces significant challenges in training, rendering and storage efficiency. Harnessing superior learning speed and real-time rendering capabilities, 3D Gaussian Splatting (3DGS) has recently demonstrated considerable potential in this field. However, 3DGS can be inefficient in terms of storage and prone to overfitting by excessively growing Gaussians, particularly with limited views. This paper proposes an efficient framework, dubbed **HiCoM**, with three key components. First, we construct a compact and robust initial 3DGS representation using a perturbation smoothing strategy. Next, we introduce a **Hi**erarchical **Co**herent **M**otion mechanism that leverages the inherent non-uniform distribution and local consistency of 3D Gaussians to swiftly and accurately learn motions across frames. Finally, we continually refine the 3DGS with additional Gaussians, which are later merged into the initial 3DGS to maintain consistency with the evolving scene. To preserve a compact representation, an equivalent number of low-opacity Gaussians that minimally impact the representation are removed before processing subsequent frames. Extensive experiments conducted on two widely used datasets show that our framework improves learning efficiency of the state-of-the-art methods by about $20\%$ and reduces the data storage by $85\%$, achieving competitive free-viewpoint video synthesis quality but with higher robustness and stability. Moreover, by parallel learning multiple frames simultaneously, our HiCoM decreases the average training wall time to $< 2$ seconds per frame with negligible performance degradation, substantially boosting real-world applicability and responsiveness. Code is avaliable at `https://github.com/gqk/HiCoM`.

38th Conference on Neural Information Processing Systems (NeurIPS 2024).

# 1   Introduction

The online reconstruction of dynamic scene from multi-view video streams is essential for advancing applications such as real-time free-viewpoint video (FVV) and virtual reality (VR), which are revolutionizing entertainment, education, and industry by providing immersive and interactive experiences. However, achieving high fidelity streamable dynamic scene poses significant challenges in training time, rendering speed, data storage and transmission efficiency.

Traditional methods for modeling and representing dynamic scenes, such as surface estimation [3], multi-sphere imaging [4], and depth mapping [5, 6] via multi-view stereo technologies, struggle with the complex geometries and varied appearances of real-world scenarios. These techniques also face limitations in capturing the detailed and dynamic nature of practical environments.

Neural Radiance Fields (NeRFs) [7, 8, 9, 10, 11, 12, 13, 14, 15, 16] have made significant breakthroughs in 3D reconstruction and novel view synthesis by mapping spatial coordinates and viewing directions to color and density using a neural network. Dynamic NeRFs [1, 17, 18, 19, 20, 21, 22, 23] integrate temporal components to capture scene changes over time. However, the high computational demands of NeRF's volume rendering framework limit their feasibility in real-time applications.

Acknowledging NeRF's limitations, researchers have introduced 3D Gaussian Splatting (3DGS) [24], an innovative method for fast 3D reconstruction and real-time rendering. In contrast to NeRF, 3DGS explicitly employs anisotropic 3D Gaussians as primitive elements to represent 3D scenes, then rasterizes these Gaussians to render images from specific viewpoints, bypassing the complex and slow volume rendering pipeline. This not only speeds up the rendering process but also enhances the quality of the synthesized views. In a similar vein to NeRF, 3DGS has been adapted for dynamic scene reconstruction: several Dynamic Gaussian Splatting works [25, 26, 27] directly expand attributes of Gaussian primitives; other efforts [28, 29, 30, 31, 32, 33, 34] focus on decoupling the dynamic scene into a base scene representation in canonical space and time-varying motion fields.

Harnessing its efficiency, 3DGS naturally supports online learning of dynamic scenes. A recent development, 3DGStream [35], introduces a pioneering framework that uses a Neural Transformation Cache derived from InstantNGP [36] to model the scene changes from previous frame to current frame, significantly reducing training time and storage requirements. However, this frame-by-frame online learning pipeline heavily relies on the quality of the initial 3DGS representation. Given that dynamic scenes are generally captured with a limited number of cameras, 3DGS can be prone to instability and overfitting when faced with sparse views. Furthermore, the number of Gaussian primitives in the initial 3DGS representation impacts the learning efficiency of subsequent frames. Despite employing multi-resolution hash encoding and lightweight MLP to alleviate inefficiency and speed up convergence, implicitly modeling the motion field still cannot fully capture critical aspects of the explicit and discrete nature of 3DGS. Although 3DGStream introduces new Gaussians to adapt to the appearance of new objects, these added Gaussians are not carried over to subsequent frames to maintain 3DGS compact. As real-world scenes evolve gradually, the discrepancies from the initial scene accumulate, necessitating more Gaussians to accurately capture these changes.

In this paper, we introduce the Hierarchical Coherent Motion (HiCoM) framework, a novel approach designed to enhance the efficiency and stability of streamable dynamic scene online reconstruction. Our HiCoM framework begins with the learning of a compact and robust initial 3DGS representation through a perturbation smoothing strategy. This ensures a reduced number of Gaussians, alleviates overfitting and establishes a foundation for consistent quality across frames. Then, we leverage the inherent non-uniform distribution and local consistency of 3D Gaussians to implement a hierarchical coherent motion mechanism. Specifically, we partition the scene into regions and recognize that only a few regions actually contain Gaussian primitives due to the non-uniform distribution of Gaussians. We explicitly model the motion within these non-empty regions, allowing Gaussians in the same region to share identical motion patterns. These regions can be further divided into smaller areas, enabling the motion of each Gaussian to be determined by the combined motions of all levels of regions it inhabits. This hierarchical coherent motion mechanism captures motions from coarse to fine granularity and requires only a minimal set of parameters, facilitates rapid convergence. The inherent structure and consistency within and between regions thus support swift learning of scene changes across frames. We also introduce additional Gaussians to better accommodate significant updates in scene content. These new Gaussians are carefully integrated into the initial 3DGS representation to ensure continuous consistency with the evolving scene. To maintain the compactness of the 3DGS,

an equivalent number of low-opacity Gaussians, which no longer significantly contribute to the scene representation, will be removed before the learning of the next frame. This continual refinement to the initial 3DGS representation ensures it remains as close as possible to the evolving scene, facilitating better subsequent learning. In addition, we introduce a parallel training strategy that enables simultaneous learning of multiple frames, significantly enhancing training efficiency with minimal impact on performance. The contributions of this paper are summarized as follows:

- We introduce the HiCoM framework for online learning of dynamic scene from multi-view video streams, featuring a perturbation smoothing strategy for robust initial 3DGS representation learning, hierarchical coherent motion mechanism for efficient motion capture, and continual refinement to adapt evolving scene updates.

- We devise a novel and concise hierarchical coherent motion mechanism that capitalizes on the inherent non-uniform distribution and local consistency of 3D Gaussians, efficiently capturing and modeling scene motions for swift and precise frame-to-frame adaptation.

- Extensive experiments demonstrate that our HiCoM framework improves learning efficiency by about $20\%$ and reduces data storage by $85\%$ compared to state-of-the-art methods. Our parallel training strategy enables learning multiple frames simultaneously, substantially decreasing the training wall time with negligible effects on overall performance, further enhancing the practicality and responsiveness of our HiCoM for real-world applications.

## 2 Related Work

### 2.1 Dynamic Gaussian Splatting

3D Gaussian Splatting [24] has quickly garnered attention for dynamic scene reconstruction due to its efficiency and flexibility. *1) Several Dynamic Gaussian Splatting works* [25, 26, 27] directly expand attributes of Gaussian primitives: 4D Gaussian Splatting [26] adds a timestamp dimension to form 4D Gaussian primitives; Dynamic 3D Gaussians [25] incorporates positions and rotations at every timestamp into the Gaussian primitives. *2) Other works* [28, 29, 30, 31, 32, 34, 33, 37] decouple the dynamic scene into base scene representation in canonical space and time-varying motion fields: Deformable 3D Gaussians [28], inspired by dynamic NeRFs [1, 22, 19, 20, 21, 18], uses a deep MLP to predict the motion of Gaussians based on their positions and the given timestamp, enabling detailed and real-time rendering of dynamic scenes; 4D-GS [29] refines this pipeline by employing a more efficient multi-resolution hex-planes and a lightweight MLP, significantly enhancing efficiency and performance, particularly in high-resolution real-time rendering scenarios; SC-GS [30] assumes that scene motions are driven by a small number of key point movements, leveraging sparse control points and dense Gaussians to distinctively represent motion and appearance in dynamic scenes, and employs a deformation MLP to predict time-varying 6 DoF transformations for each control point, which are then locally interpolated through learned weights to create a motion field that influences the dense Gaussians, ensuring coherent and dynamic scene rendering.

### 2.2 Online Learning of Streamable Dynamic Scene

Compared to the offline learning from fully captured multi-view videos, on-the-fly learning, *i.e.*, frame-by-frame training, for constructing streamable dynamic scenes presents more challenges. Given the success of NeRFs [7, 8, 9, 36, 20, 19] in addressing both static and dynamic scenes, there has been a growing interest in adapting and refining these techniques to solve the challenges of streamable dynamic scenes. StreamRF [2] tackles this by employing an incremental learning paradigm, where per-frame differences are modeled to efficiently adapt to changes, significantly speeding up the training process and reducing storage overhead than traditional methods. NeRFPlayer [38], on the other hand, decomposes the 4D spatiotemporal space based on temporal characteristics, optimizing for both speed and compactness in model representation, thus enabling fast reconstruction and streamable rendering. ReRF [39] extends these capabilities by modeling the residual information between adjacent timestamps, which allows for real-time free-viewpoint video (FVV) rendering of long-duration dynamic scenes with high fidelity. The emergence of 3D Gaussian Splatting (3DGS) has brought significant advancements in training efficiency and rendering speeds, making it exceptionally suited for addressing streamable dynamic scenes. Building on superior 3DGS and inspired by previous NeRF works [2, 36], 3DGStream [35] introduces an efficient framework that

uses a Neural Transformation Cache to manage the transformations of 3D Gaussians, drastically reducing the training time and storage requirements. This method also features an adaptive addition strategy for 3D Gaussians to better handle emerging objects in dynamic scenes.

## 3 Preliminaries

3D Gaussian Splatting explicitly represents scenes using anisotropic 3D Gaussian primitives, each defined by a mean vector $\boldsymbol{\mu}$ and covariance matrix $\boldsymbol{\Sigma}$, which respectively characterize the central position and geometric shape of the Gaussian in world space, mathematically formulated as:

$$G(\mathbf{x}) = e^{-\frac{1}{2}(\mathbf{x}-\boldsymbol{\mu})^T \boldsymbol{\Sigma}^{-1}(\mathbf{x}-\boldsymbol{\mu})}. \tag{1}$$

The covariance matrix $\boldsymbol{\Sigma}$ is decomposed into a scaling matrix $\mathbf{S}$ and a rotation matrix $\mathbf{R}$ to ensure physical meaning and facilitate optimization:

$$\boldsymbol{\Sigma} = \mathbf{R}\mathbf{S}\mathbf{S}^T\mathbf{R}^T, \tag{2}$$

where $\mathbf{S} = \mathrm{diag}(s_x, s_y, s_z) \in \mathbb{R}^3$ and $\mathbf{R} \in SO(3)$ are parameterized as a 3D scaling vector $\mathbf{s}$ and rotation quaternion $\mathbf{q}$, respectively. Gaussian primitive is enriched with color and opacity, represented by spherical harmonic coefficients $\mathbf{h}$ and a scalar $\alpha$, respectively.

To render a novel viewpoint, Gaussian primitives are projected onto the camera plane using the view projection matrix $\mathbf{W}$ and the Jacobian $\mathbf{J}$ of the affine approximation of the projective transformation, the covariance matrix $\boldsymbol{\Sigma}'$ in camera coordinates given as follows:

$$\boldsymbol{\Sigma}' = \mathbf{J}\mathbf{W}\boldsymbol{\Sigma}\mathbf{W}^T\mathbf{J}^T. \tag{3}$$

Rendering is performed by blending the contributions of $N$ overlapping Gaussian primitives at each pixel, taking into account their depth-ordering to ensure correct compositing, expressed as:

$$C = \sum_{i \in N} \mathbf{c}_i \alpha_i \prod_{j=1}^{i-1}(1 - \alpha_j). \tag{4}$$

where $\mathbf{c}_i$, $\alpha_i$ represents the color and blending weight of the $i^{th}$ Gaussian, respectively.

The training of 3D Gaussian Splatting alternates between parameter optimization and density control. Parameter optimization is supervised by the $\mathcal{L}_1$ loss and D-SSIM term:

$$\mathcal{L} = (1 - \lambda)\mathcal{L}_1 + \lambda\mathcal{L}_{\text{D-SSIM}} \tag{5}$$

where $\lambda$ is typically set to 0.2. Meanwhile, density control manages Gaussian cloning and splitting to address over-reconstruction and under-reconstruction.

## 4 Methodology

The online reconstruction of streamable dynamic scenes follows a sequential and iterative pipeline, crucial for applications requiring real-time rendering and interaction. Initially, a 3D Gaussian Splatting (3DGS) representation of the scene is constructed from the first frame of synchronized multi-view video streams. As each new frame is captured, this representation is updated and refined to align with the latest scene. This process demands a high-quality scene representation at every timestamp to ensure visual fidelity, while also necessitating efficiency in learning, storage, and transmission to make it feasible for real-world applications.

To achieve these goals, our framework (Figure 2) introduces three key designs: First, we develop a compact and robust initial 3DGS representation using a perturbation smoothing strategy, ensuring a stable foundation for subsequent frame learning (Sec. 4.1). Second, we employ a hierarchical coherent motion mechanism, leveraging the inherent non-uniform distribution and local consistency of 3D Gaussians to model and rapidly learn scene motion from the previous frame to the current frame (Sec. 4.2). Third, we implement continual refinement strategies to adapt to scene content updates missed by the motion mechanism, maintaining fidelity and accuracy as the scene evolves (Sec. 4.3). Importantly, we develop a parallel training strategy that enables simultaneous learning of multiple frames (Sec. 4.4), significantly enhancing the practicality and responsiveness of our framework.

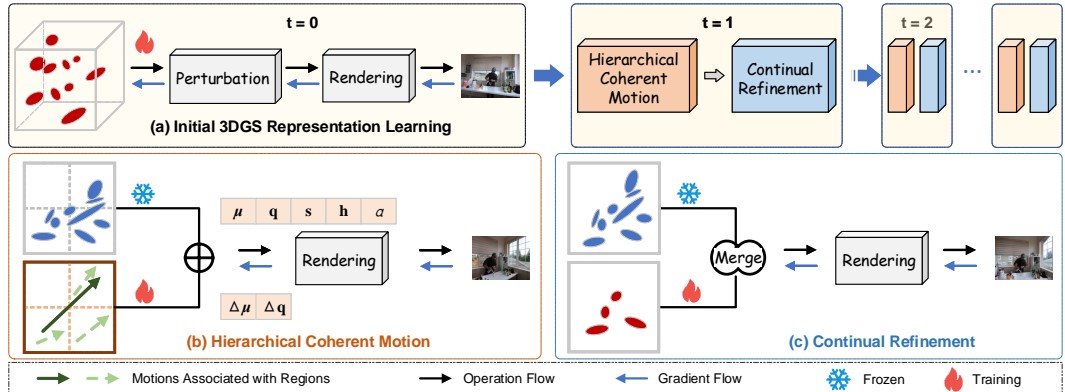

Figure 2: **Illustration of our HiCoM framework.** We first construct a compact and robust initial 3DGS representation from the first frame of the video streams with the perturbation smoothing strategy **(a)**. Then, we learn each subsequent frame based on the previous 3DGS with our proposed Hierarchical Coherent Motion mechanism **(b)** and Continual Refinement **(c)** with additional new Gaussians. This learning process continues until the final frame.

## 4.1 Initial 3DGS Representation Learning

The initial 3DGS representation forms the cornerstone of the entire dynamic scene reconstruction process, laying the groundwork for precise and efficient frame-by-frame online learning.

In typical setups for capturing dynamic scenes, the number of cameras is often limited, contrasting sharply with static scenes that usually involve hundreds of multi-view images. This limitation can lead to instability and potential overfitting to training views during optimization. Moreover, if not adequately controlled, the number of Gaussians could grow unchecked, slowing down the rendering process and convergence speed in the learning of subsequent frames.

To combat overfitting, a prevalent issue in neural network training, various strategies such as dropout [40] and data augmentation [41, 42, 43] have been proposed. Drawing from these insights, we discover that a simple perturbation smoothing strategy helps mitigating this issue in 3DGS, which has proven also effective in NeRF [44]. As **Figure 2 (a)**, we add small gaussian noise to the position attribute of 3D Gaussian during training, creating the random perturbed position:

$$\boldsymbol{\mu}_p = \boldsymbol{\mu} + \lambda_{noise}\boldsymbol{\epsilon} \quad \text{where} \quad \boldsymbol{\epsilon} \sim \mathcal{N}(0, 1), \tag{6}$$

thereby preventing the model from becoming too closely fitted to the limited training views. Random perturbation not only stabilizes the learning process but also moderates the growth of 3D Gaussians. As a result, by the time the model converges, the number of Gaussians is significantly reduced, speeding up the convergence and enhancing the overall efficiency in subsequent frame learning.

## 4.2 Hierarchical Coherent Motion

Follow the online learning pipeline, we continuously fetch the latest views from the video streams and reconstruct the current scene representation. A straightforward method would be to use these new images to fine-tune the previous 3DGS. However, this naive approach is still insufficient in terms of learning speed and requires storing and transmitting a significant amount of data.

Although recent efforts [28, 29, 35] have demonstrated the feasibility of using neural networks to predict the motion of each 3D Gaussian, showing promising results in adapting to scene changes, the implicit modeling of the motion field fails to align well with the explicit and discrete nature of 3DGS. In contrast, we explicitly model the motion field through hierarchical coherent motion mechanism, allowing us to directly capture and adapt to scene dynamics in a structured and efficient manner.

As illustrated in **Figure 2 (b)**, the scene is first divided into regions with the same size. and we can determine the region in which each 3D Gaussian is primarily located using the following formula:

$$\mathbf{r} = \left\{ \left\lfloor \frac{\boldsymbol{\mu}}{\mathbf{e}} + 0.5 \right\rfloor \right\} \cdot \mathbf{e}, \tag{7}$$

where $\mathbf{r}$ denotes the center coordinates of the region, $\boldsymbol{\mu}$ is the center position of the 3D Gaussian, and $\mathbf{e}$ represents the size of region.

During optimization, 3D Gaussians naturally exhibit a non-uniform distribution, with areas of complex geometry and texture densely populated by smaller Gaussians, while simpler and smoother regions sparsely covered by larger Gaussians, resulting in a significant number of regions remaining empty. For those regions that contain Gaussians, we assign translation and rotation attributes, respectively parameterized by a 3D vector $\Delta\boldsymbol{\mu} \in \mathbb{R}^3$ and a quaternion $\Delta\mathbf{q} \in \mathbb{R}^4$. These motion attributes are shared by the Gaussians within the region, ensuring local consistency.

However, motion within smaller subregions might exhibit subtle differences due to more localized dynamics. To accommodate these variations, we further partition the regions and establish distinct motion parameters for these smaller areas. Consequently, the motion of each Gaussian is determined by the cumulative of motion from multiple hierarchical levels to which it belongs:

$$\Delta\boldsymbol{\mu}_g = \sum_{l=1}^{L} \Delta\boldsymbol{\mu}^l, \quad \Delta\mathbf{q}_g = \sum_{l=1}^{L} \Delta\mathbf{q}^l \tag{8}$$

where $\Delta\boldsymbol{\mu}_g \in \mathbb{R}^3$ and $\Delta\mathbf{q}_g \in \mathbb{R}^4$ represent the composed motion of Gaussian, $\Delta\boldsymbol{\mu}^l$ indicates the motion contribution from the $l^{th}$ level of regions, and $L$ is the total number of motion levels involved.

This hierarchical coherent motion mechanism allows us to capture Gaussians' movements ranging from coarse to fine, ensuring a more detailed and accurate representation of dynamic changes across frames, as well as coherent motion between adjacent regions. By sharing motion attributes among multiple Gaussians, we decrease the number of individual computations required and the number of parameters we need to optimize, helping the learning converge faster and more reliably.

### 4.3 Continual Refinement

A 3DGS representation that is more closely aligned with the most recent scene can be obtained after applying the learned motion. However, this motion mechanism, focusing primarily on modest shifts to the positions and rotations of Gaussians from previous 3DGS, might not capture all the finer details of scene changes or adapt swiftly enough to significant scene content updates. This gap can be strategically bridged by adding new Gaussians in appropriate areas.

Specifically, during motion learning, some regions accumulate significant gradients indicating substantial discrepancies between the learned and actual scene. Following established methods [24, 35], we clone Gaussians in these regions that have accumulated gradients exceeding a predefined threshold. As shown in **Figure 2 (c)**, the newly cloned Gaussians are then subjected to further optimization and density control to better align with the most recent scene.

As real-world dynamic scenes typically exhibit coherent changes, newly added Gaussians tend to stay relevant across subsequent frames, making it impractical to simply discard them. Thus, we retain these Gaussians, integrating them into the initial 3DGS to form the basis for learning the next frame. However, consistently merging new Gaussians in each frame can lead to a steady increase in the number of Gaussians, potentially slowing down rendering speeds, delaying convergence, and prolonging learning times. Therefore, we selectively remove an equivalent number of low-opacity Gaussians that minimally impact the overall visual integrity of the scene [45], to maintain the compactness and efficiency of 3DGS representation without compromising scene fidelity. Simultaneously, this continual refinement allows us to address and enhance regions of the 3DGS that may have been underrepresented or insufficiently modeled in earlier learning stages, solidifying a comprehensive and adaptive representation as the scene evolves.

### 4.4 Parallel Training

Typically, our framework works in a sequential frame-by-frame pipeline. To further boost efficiency and responsiveness for practical applications, we draw inspiration from video encoding techniques where reference frames (such as I-frames in H.264/AVC) are used to predict multiple subsequent frames, leveraging temporal redundancy and reducing computational overload.

Considering that differences between consecutive frames are generally minor, we choose the 3DGS of the frame $t$ as the reference to simultaneously learn frames $\{t+1, \cdots, t+k\}$. After processing

Table 1: **Quantitative comparison results**. "Storage" is reported without/with the initial frame. The method with † is reproduced by us. StreamRF metrics on Meet Room only cover the discussion scene. All experiments use original datasets without additional undistortion. Please also see Tables 6 and 7.

| Method | N3DV | | | | Meet Room | | | |
|---|---|---|---|---|---|---|---|---|
| | PSNR (dB ↑) | Storage (MB ↓) | Train (s ↓) | Render (FPS ↑) | PSNR (dB ↑) | Storage (MB ↓) | Train (s ↓) | Render (FPS ↑) |
| Naive 3DGS | 30.69 | 95.8 | 337 | 217 | 26.37 | 46.9 | 152 | 275 |
| StreamRF [2] | 30.68 | 17.7/31.4 | 10.0 | 13 | 26.72 | 5.7/9.0 | 6.8 | 15 |
| 3DGStream† [35] | 30.15 | 7.6/7.8 | 8.2 | 210 | 25.96 | 4.0/4.1 | 4.7 | 212 |
| **HiCoM** (ours) | 31.17 | 0.7/0.9 | 6.7 | 274 | 26.73 | 0.4/0.6 | 3.9 | 284 |
| **HiCoM-P4** (ours) | 31.00 | 0.7/0.9 | 1.7 | 274 | 26.25 | 0.4/0.6 | 1.0 | 284 |

these frames, the 3DGS of the frame $t + k$ becomes the new reference for learning the next $k$ frames. This parallel training strategy significantly reduce the average wall time cost per frame.

## 5 Experiment

### 5.1 Experimental Setup

For all of our experiments in this paper, we utilize two widely-used public datasets as follows.
• **Neural 3D Video (N3DV)** [1] dataset comprises six dynamic indoor scenes featuring varying illuminations, view-dependent effects, and substantial volumetric details. Videos were captured at a resolution of $2704 \times 2078$ and a frame rate of 30 FPS. Video counts per scene range from 18 to 21. We follow prior works [2, 35, 29] to downsample the original video resolution by a factor of two.
• **Meet Room** [2] dataset was recorded with a multi-view system consisting of 13 synchronized Azure Kinect cameras, covering three different indoor scenes. The resolution of captured videos is $1280 \times 720$ and the frame rate is 30 FPS too.

For both datasets, the video captured by the center reference camera is used for testing, while other videos are employed for training. Both datasets already include camera pose information, so we only generate initial point clouds with COLMAP [46] following 4D-GS [29]. Most scenes last only 10 seconds, thus, we restrict our experimentation to the first 300 frames to align with prior works.

We evaluate methods based on four metrics, across all 300 frames: *1) video synthesis quality* of the test views, assessed by the mean PSNR; *2) average training time per frame*; *3) rendering speed*, assessed by the frame rate of the synthesized video; *4) storage and transmission efficiency*, evaluated by the average size of the representation parameters per frame.

**Implementation**. Our initial frame learning employs the standard 3DGS with a fixed $\lambda_{noise}$=0.01, training for 15k and 10k steps on two datasets respectively, with splitting halted at the $5k^{th}$ step. The number of regions at the smallest level is determined by dividing the number of Gaussians $n$ in the initial 3DGS by the maximum number of Gaussians $m$=5 per region. The number of regions at each subsequent larger level is determined by further dividing by $2^3$, and so forth, with only regions containing Gaussians having motion parameters, which are initialized at $0.6\times$ of the previous frame's motion parameters. Each new frame undergoes 100 motion training steps, with an additional 100 steps after new Gaussians are added. We implement 3DGStream [35] in the same codebase as ours following its paper, and run all experiments 3 times on RTX 4090 GPUs, the mean metrics are reported. More details are provided in the **Appendix**.

### 5.2 Experimental Results

**Quantitative Benchmark** We conduct a comprehensive quantitative comparison with state-of-the-art online methods, StreamRF [2] and 3DGStream [35], presenting results in Table 1. The Naive 3DGS serves as a baseline, training from scratch frame-by-frame with standard 3DGS method. The data of StreamRF are referenced from the 3DGStream paper, with training time and rendering speed estimated based on the compute capabilities of RTX 3090 and 4090 GPUs. Our HiCoM consistently achieves competitive video synthesis quality (PSNR), while significantly outpacing the

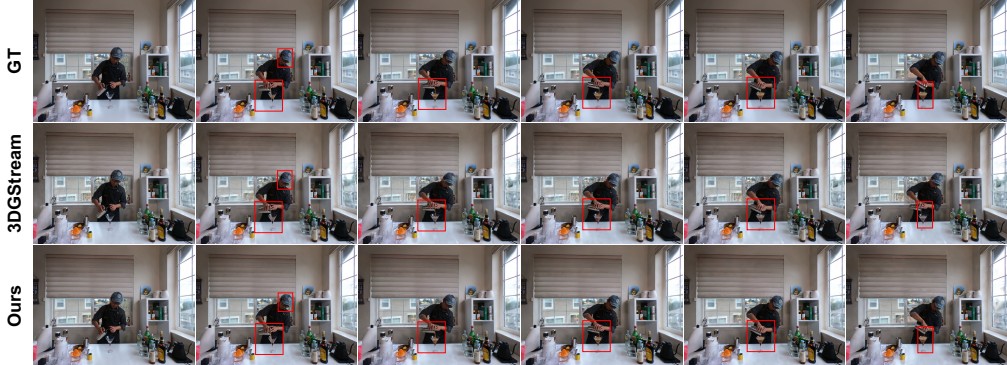

Figure 3: **Qualitative results of Coffee Martini scene.** Frames shown are the $1^{st}$, $61^{st}$, $121^{st}$, $181^{st}$, $241^{st}$, and $300^{th}$ from the test video. Red boxes highlight areas with significant temporal motions. Our method achieves temporal coherence more closely matching the ground truth (GT).

two counterparts in training speed, with an average per-frame learning time (include the initial frame) reduced by more than 17% compared to 3DGStream on both datasets. This advantage becomes even more pronounced in complex scenes, such as Coffee Martini and Flame Salmon, where the learning speed improvement exceeds 20%, primarily due to our method's rapid convergence speed. Both our HiCoM and 3DGStream obtain rendering speeds over 200 fps, demonstrating real-time rendering capability. In terms of storage and transmission efficiency, our approach substantially surpasses the two competitors, requiring less than 10% of their average per-frame storage space after the initial frame. This is mainly due to two factors: firstly, our motion parameters are shared among multiple 3D Gaussians within local regions; secondly, the inherent non-uniform distribution of the 3D Gaussians results in most areas being empty, thus requiring far fewer motion parameters. Due to the reduced amount of information needing transmission, our method is more advantageous for streamable dynamic scenes. In the **Appendix** (Tables 6, 7), we provide results for each scene and results of several state-of-the-art offline methods, offering a more detailed quantitative comparison. We also include experimental results on two additional datasets, PanopticSports [25] and ParticleNeRF [47], in the **Appendix**. On the PanopticSports dataset, which features large-scale motion, our method achieves better performance than the state-of-the-art Dynamic3DGS [25] method, while on the synthetic ParticleNeRF dataset, our HiCoM demonstrates good results as well. These additional experiments further validate the generalization capability of our approach across diverse datasets.

**Qualitative Analysis**   Our HiCoM framework inherits the advantages of 3D Gaussian Splatting in representing static scenes but focuses more on the changes in dynamic scenes over time. As shown in Figure 3, the scene features a person making a drink, with his head, hands, and the liquid in the cup being the main dynamic areas. We select six frames, arranged in chronological order, from all 300 frames to analyze the rendering quality. It can be observed that our method produces results that are much closer to the ground truth in these dynamic areas. Furthermore, our HiCoM exhibits more coherent temporal transitions because, in addition to updating the initial Gaussian positions and rotations, we carry over newly added Gaussians to the subsequent frames instead of discarding them. This results in smoother and more natural transitions over time, such as the liquid gradually flowing into the glass. More qualitative results for other scenes are provided in the **Appendix**.

### 5.3   Ablation Study

As shown in Table 2, we ablate three key components of our HiCoM framework. In the $2^{nd}$ row, we do not use noise perturbation, *i.e.*, the initial frame learning falls back to standard 3DGS training. This causes many small Gaussians to split during the initial learning to fit the training views, resulting in a slight decrease in PSNR on the test views. By incorporating noise perturbation, our HiCoM reduces the number of Gaussians while making them larger, each contributing to the rendering of more pixels. Consequently, the training time was slightly longer ($1^{st}$ row *vs.* $2^{nd}$ row). In the $3^{rd}$ row, we disable motion learning, using the first 100 steps solely for gradient accumulation to determine which Gaussians to clone. This resulted in a significant performance drop, particularly in the relatively simple scenes Flame Steak and Discussion, indicating that this component is crucial for

Table 2: **Ablation on main components of our HiCoM framework**. Coffee Martini and Flame Steak are from the N3DV dataset, while Discussion is from the Meet Room dataset.

| Pertubation | Motion | Refinement | Coffee Martini | | Flame Steak | | Discussion | |
|:---:|:---:|:---:|:---:|:---:|:---:|:---:|:---:|:---:|
| | | | PSNR (dB ↑) | Train (s ↓) | PSNR (dB ↑) | Train (s ↓) | PSNR (dB ↑) | Train (s ↓) |
| ✓ | ✓ | ✓ | **28.04** | 7.0 | **32.87** | 6.6 | **26.69** | 3.9 |
| ✗ | ✓ | ✓ | 26.61 | **6.3** | 32.52 | **5.6** | 26.19 | 3.6 |
| ✓ | ✗ | ✓ | 25.71 | 6.4 | 27.33 | 5.9 | 21.61 | **3.4** |
| ✓ | ✓ | ✗ | 27.87 | 7.6 | 31.73 | 6.7 | 26.51 | 4.2 |

Table 3: **Ablation on convergence.** "$E_m$" and "$E_r$" are number of motion learning steps and continual refinement steps, repectively.

| $E_m$ | $E_r$ | N3DV | | Meet Room | |
|:---:|:---:|:---:|:---:|:---:|:---:|
| | | PSNR (dB ↑) | Train (s ↓) | PSNR (dB ↑) | Train (s ↓) |
| 100 | 100 | **31.17** | 6.7 | 26.73 | 3.9 |
| 50 | 100 | 30.78 | **5.2** | 26.51 | **3.0** |
| 150 | 100 | 31.16 | 8.2 | **26.77** | 4.8 |
| 100 | 50 | 30.94 | 5.3 | **26.77** | 3.1 |
| 100 | 150 | 31.10 | 8.1 | 26.70 | 4.6 |

the overall framework. In the $4^{th}$ row, we omit the refinement, extending motion learning instead. This leads to a slight performance decrease, suggesting that the additional Gaussians can further compensate for the intense and detailed scene changes that the motion learning might not capture.

A notable characteristic of our HiCoM is the explicit representation of motion field, which aligns well with the discrete, explicit nature of 3DGS. Our HiCoM requires very few motion parameters, reducing the complexity of motion representation and learning, theoretically accelerating the convergence speed. According to ablation results presented in Table 3, increasing the motion learning steps from 50 to 100 ($2^{nd}$ row *vs.* $1^{st}$ row) still provides significant performance gains, *i.e.*, 0.39 and 0.22 dB on two datasets, respectively. However, learning an additional 50 steps beyond 100 ($1^{st}$ row *vs.* $3^{rd}$ row) yields only minimal improvements , indicating that the model has mostly converged by 100 steps. Similar conclusions can be drawn for learning newly added Gaussians. Ultimately, a total of 200 learning steps is sufficient to achieve optimal results.

Our hierarchical coherent motion incorporates multiple levels of motion, with different levels corresponding to varying sizes of regions and reflecting different granularities of motion. In Table 4, we present ablation studies on motion levels. We use three motion levels by default (the $5^{th}$ row), refer to as "fine", "medium" and "coarse", respectively. As the results show, using only the "fine" level already achieves good performance. Adding the "medium" and "coarse" levels provides additional performance gains, *e.g.*, the PSNR of Coffee Martini and Discussion scenes respectively boosts 0.25 dB and 0.59 dB, but the improvements diminish as the number of levels increases, indicating that our HiCoM is effective and a few motion levels are sufficient to achieve good results. It can also be observed that fewer motion levels and parameters do not necessarily imply shorter training time. This is because insufficient motion learning may lead to the generation of more Gaussians during continual refinement stage, slightly increasing the overall training time.

## 5.4 Parallel Training

We conduct experiments on three representative scenes from two datasets to validate the parallel learning. As shown in Table 5, when the number of parallel frames is relatively small, such as 2 and 4, the performance remains largely unchanged, with some scenes even exhibiting improved PSNR. However, as the number of parallel frames increases, the disparities between the learning and the reference frames grow, leading to a decline in performance. Specifically, when the number of parallel frames reaches 16, there is a noticeable PSNR drop, *i.e.*, 0.18, 1.82 and 1.32 dB, respectively.

Table 4: **Ablation on motion levels.** Coffee Martini and Flame Steak are from the N3DV dataset, while Discussion is from the Meet Room dataset. The three motion levels correspond to regions of different sizes: "fine" for smaller regions, "medium" for intermediate regions, and "coarse" for larger regions. Rows show results for different combinations of these levels.

| Motion Levels | Coffee Martini | | Flame Steak | | Discussion | |
|---|---|---|---|---|---|---|
| | PSNR (dB ↑) | Train (s ↓) | PSNR (dB ↑) | Train (s ↓) | PSNR (dB ↑) | Train (s ↓) |
| coarse | 26.67 | 6.77 | 30.28 | **6.26** | 22.89 | 3.83 |
| medium | 27.24 | **6.60** | 32.14 | **6.26** | 24.91 | 3.80 |
| fine | 27.79 | 6.65 | 32.76 | 6.33 | 26.10 | **3.72** |
| fine + medium | 27.94 | 6.77 | **32.88** | 6.37 | 26.61 | 3.77 |
| fine + medium + coarse | **28.04** | 7.02 | 32.87 | 6.62 | **26.69** | 3.89 |

Table 5: **Parallel Training Performance** on three scenes from N3DV and MeetRoom datasets.

| #Parallel Frames | Coffee Martini | | Flame Steak | | Discussion | |
|---|---|---|---|---|---|---|
| | PSNR (dB ↑) | Train (s ↓) | PSNR (dB ↑) | Train (s ↓) | PSNR (dB ↑) | Train (s ↓) |
| Non-Parallel | 28.04 | 7.02 | **32.87** | 6.62 | **26.69** | 3.89 |
| 2 | **28.20** | 3.51 | 32.46 | 3.31 | 26.66 | 1.95 |
| 4 | 28.19 | 1.76 | 32.13 | 1.66 | 26.39 | 0.97 |
| 8 | 28.06 | 0.88 | 31.63 | 0.83 | 25.89 | 0.49 |
| 16 | 27.86 | 0.44 | 31.05 | 0.42 | 25.37 | 0.25 |

While parallel learning does not reduce the actual GPU computation time, it significantly reduces the wall time, making the system more practical for real-world applications. For example, with 4-frame parallel learning, we can reduce the average learning time per frame to one-fourth of the original time while maintaining nearly the same quality. This means we can learn one frame in less than 2 seconds on average, greatly enhancing the efficiency and responsiveness of the system.

## 6 Conclusion

This paper introduces HiCoM, a framework designed to tackle the challenges in online reconstruction of streamable dynamic scenes from multi-view videos. HiCoM reduces the number of 3D Gaussians for a more compact representation through the perturbation smoothing strategy, enhancing the robustness of initial frame learning. HiCoM distinguishes with its hierarchical coherent motion mechanism that explicitly represents motion between adjacent frames with minimal parameters, aligning well with the discrete and explicit nature of 3D Gaussian Splatting. This mechanism efficiently captures motion at different granularities, resulting in faster convergence and substantially reduced data storage. Additionally, HiCoM continuously refines the representation to adapt to evolving scene, facilitating better subsequent learning. Experimental results show that HiCoM boosts training efficiency by about $20\%$ and reduces data storage by $85\%$ compared to state-of-the-art methods. Furthermore, the parallel training significantly reduces wall time cost with negligible performance degradation, elevating real-world applicability and responsiveness.

**Limitations.** Despite our HiCoM's significant improvements, the role of the initial 3DGS representation remains crucial in the online learning pipeline, which is still not fully addressed. Future work could explore integrating advanced 3D Gaussian Splatting techniques to enhance the initial frame learning, which, combined with our hierarchical coherent motion mechanism, could further improve training efficiency while also reducing storage and transmission overhead. During online learning, reconstruction quality may decline over time due to error accumulation—a common issue across many online learning methods and one that requires further research to effectively address. Additionally, our experiments are conducted solely on indoor scenes, so the generalization of our method to outdoor or more complex environments has not been verified and requires further validation.

## Acknowledgements

This work was supported in part by the National Key R&D Program of China (No. 2022ZD0118201), the Shenzhen Medical Research Funds in China (No. B2302037), Natural Science Foundation of China (No. 61972217, 32071459, 62176249, 62006133, 62271465), Guangdong Provincial Key Laboratory of Ultra High Definition Immersive Media Technology (No. 2024B1212010006), and AI for Science (AI4S)-Preferred Program, Peking University Shenzhen Graduate School, China.

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

# Appendix

This appendix provides additional material to supplement the main text. We will first introduce more implementation details in Sec. A. Then we provide some additional experimental results in Sec. B.

## A  More Implementation Details

Our code is primarily based on the open-source codes of 3DGS [24] and 4D-GS [29]. We keep the 4D-GS setting that does not reset opacity in the initial 3DGS learning because we observed that resetting opacity leads to the removal of larger Gaussians, which is helpful when the number of Gaussians already significantly reduced by our noise perturbation smoothing mechanism. The hyperparameter $\lambda_{noise}$ was empirically selected from the range of $\{0.001, 0.01, 0.1\}$. Among these values, 0.01 consistently performed the best across multiple scenes, hence we set it to 0.01 in all experiments. After applying the learned motion to 3DGS, we determine the Gaussians to be cloned based on the accumulated gradients during motion learning, which is similar to standard 3DGS. The cloned Gaussians undergo alternating optimization and densification, with densification occurring every 40 steps. We found that this parameter is not sensitive and can be freely set between 20 and 50. The removal of low-opacity Gaussians happens before learning the next frame, so these Gaussians still contribute to the representation of the current frame.

At the time of conducting this work, the code[1] released by the authors of 3DGStream still demonstrated instability and encountered issues during execution. Therefore, we reported the experimental results based on the code we implemented. Although, we attempted to faithfully reproduce the 3DGStream as described in its paper, inconsistencies remain in our implementation and experimental setup. The main difference lies in the dataset processing. We employ the original dataset following 4D-GS code, whereas 3DGStream utilizes processed datasets with modified resolutions and camera poses, resulting in about 50,000 fewer pixels per view image of N3DV [1] dataset. The spherical harmonics (SH) order is set to 1 as per the paper, but we do not perform SH rotation because it increases the computational overhead without noticeable performance gains. Additionally, 3DGStream requires scene boundary being manually setup, whereas we automatically compute them after removing some outlier Gaussians. These disparities lead to marginally lower metrics compared to those reported in the 3DGStream paper, but our results align closely with other offline methods using the same dataset.

## B  Additional Experimental Results

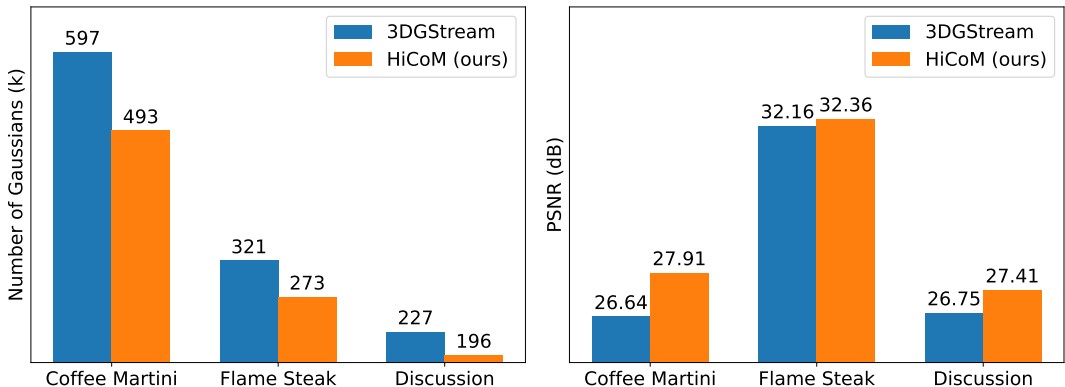

Figure 4: **Comparison of initial 3DGS.** 3DGStream utlizes the standard 3DGS training, our HiCoM additionally incorporates noise perturbation.

---

[1]https://github.com/SJoJoK/3DGStream

Table 6: **Per-scene quantitative results on the N3DV dataset.** The offline and online methods are separated into upper and lower sections, respectively. The per-frame metrics "Storage" and "Train" are averaged over 300 frames, with those for offline methods [20, 38, 26, 29, 48, 34] estimated based on data reported in papers [29, 48, 34]. The data for Dynamic 3DGS are referenced from [48].

| Method | Coffee Martini | | | Cook Spinach | | | Cut Beef | | |
|---|---|---|---|---|---|---|---|---|---|
| | PSNR (dB ↑) | Storage (MB ↓) | Train (s ↓) | PSNR (dB ↑) | Storage (MB ↓) | Train (s ↓) | PSNR (dB ↑) | Storage (MB ↓) | Train (s ↓) |
| KPlanes [20] | 29.99 | 1.0 | 14.4 | 32.60 | 1.0 | 14.4 | 31.82 | 1.0 | 14.4 |
| NeRFPlayer [38] | 31.53 | 18.4 | 48.0 | 30.56 | 18.4 | 48.0 | 29.35 | 18.4 | 48.0 |
| 4DGS [26] | 28.33 | 29.0 | 76.0 | 32.93 | 29.0 | 76.0 | 33.85 | 29.0 | 76.0 |
| 4D-GS [29] | 27.34 | 0.3 | 5.3 | 32.46 | 0.3 | 5.3 | 32.90 | 0.3 | 5.3 |
| Spacetime-GS [48] | 28.61 | 0.7 | 28.0 | 33.18 | 0.7 | 28.0 | 33.52 | 0.7 | 28.0 |
| E-D3DGS [34] | 29.33 | 0.5 | 43.0 | 33.19 | 0.5 | 43.0 | 33.25 | 0.5 | 43.0 |
| Dynamic 3DGS [25] | 26.49 | 9.2 | 560 | 32.97 | 9.2 | 560 | 30.72 | 9.2 | 560 |
| 3DGStream [35] | 26.73 | 7.6 | 9.26 | 31.38 | 7.6 | 7.79 | 31.36 | 7.6 | 7.73 |
| **HiCoM** (ours) | 28.04 | 0.8 | 7.02 | 32.45 | 0.6 | 6.47 | 32.72 | 0.6 | 6.56 |
| **HiCoM-P4** (ours) | 28.19 | 0.8 | 1.76 | 32.34 | 0.6 | 1.62 | 32.36 | 0.6 | 1.64 |

| Method | Flame Salmon | | | Flame Steak | | | Sear Steak | | |
|---|---|---|---|---|---|---|---|---|---|
| | PSNR (dB ↑) | Storage (MB ↓) | Train (s ↓) | PSNR (dB ↑) | Storage (MB ↓) | Train (s ↓) | PSNR (dB ↑) | Storage (MB ↓) | Train (s ↓) |
| KPlanes [20] | 30.44 | 1.0 | 14.4 | 32.38 | 1.0 | 14.4 | 32.52 | 1.0 | 14.4 |
| NeRFPlayer [38] | 31.65 | 18.4 | 48.0 | 31.93 | 18.4 | 48.0 | 29.12 | 18.4 | 48.0 |
| 4DGS [26] | 29.38 | 29.0 | 76.0 | 34.03 | 29.0 | 76.0 | 33.51 | 29.0 | 76.0 |
| 4D-GS [29] | 29.20 | 0.3 | 5.3 | 32.51 | 0.3 | 5.3 | 32.49 | 0.3 | 5.3 |
| Spacetime-GS [48] | 29.48 | 0.7 | 28.0 | 33.64 | 0.7 | 28.0 | 33.89 | 0.7 | 28.0 |
| E-D3DGS [34] | 29.72 | 0.5 | 43.0 | 33.55 | 0.5 | 43.0 | 33.55 | 0.5 | 43.0 |
| Dynamic 3DGS [25] | 26.92 | 9.2 | 560 | 33.24 | 9.2 | 560 | 33.68 | 9.2 | 560 |
| 3DGStream [35] | 27.45 | 7.6 | 9.14 | 31.56 | 7.6 | 7.77 | 32.44 | 7.6 | 7.72 |
| **HiCoM** (ours) | 28.37 | 0.9 | 7.06 | 32.87 | 0.6 | 6.62 | 32.57 | 0.6 | 6.40 |
| **HiCoM-P4** (ours) | 28.54 | 0.9 | 1.77 | 32.13 | 0.6 | 1.66 | 32.42 | 0.6 | 1.60 |

Table 7: **Per-scene quantitative results on the Meet Room dataset.** Rows marked with "*" indicate experiments conducted on the undistorted version of the dataset.

| Method | Discussion | | | Trimming | | | VR Headset | | |
|---|---|---|---|---|---|---|---|---|---|
| | PSNR (dB ↑) | Storage (MB ↓) | Train (s ↓) | PSNR (dB ↑) | Storage (MB ↓) | Train (s ↓) | PSNR (dB ↑) | Storage (MB ↓) | Train (s ↓) |
| 3DGStream [35] | 25.94 | 4.0 | 4.78 | 26.18 | 4.0 | 4.69 | 25.75 | 4.0 | 4.70 |
| **HiCoM** (ours) | 26.69 | 0.6 | 3.89 | 26.99 | 0.4 | 3.85 | 26.51 | 0.3 | 3.88 |
| **HiCoM-P4** (ours) | 26.39 | 0.6 | 0.97 | 26.38 | 0.4 | 0.96 | 25.98 | 0.3 | 0.97 |
| 3DGStream* [35] | 30.06 | 4.0 | 4.40 | 28.82 | 4.0 | 4.34 | 29.26 | 4.0 | 4.39 |
| **HiCoM*** (ours) | 29.61 | 0.4 | 3.65 | 29.64 | 0.4 | 3.65 | 29.86 | 0.3 | 3.76 |

## B.1 Inital 3DGS Comparison

The initial 3DGS representation is crucial for the overall performance of subsequent online learning. In Figure 4, we compare 3DGStream that utilizes the standard 3DGS training with our HiCoM that additionally incorporates noise perturbation. It is evident that our noise perturbation successfully reduces the number of 3D Gaussians in the initial representation, while slightly enhancing the PSNR metric. This subtle improvement highlights the superiority of our method. However, as seen in the ablation experiments detailed in the main text, when our method employs the same standard 3DGS training for the initial frame, it only achieves a PSNR comparable to that of 3DGStream. This indicates the importance of fully leveraging the benefits of latest advancements within the 3DGS domain to further improve the initial 3DGS representation.

Table 8: **Performance stability across three runs with the same random seed.** Results are reported as the mean and standard deviation from three runs on the N3DV dataset with seed 0.

| Method | Coffee Martini | Cook Spinach | Cut Beef | Flame Salmon | Flame Steak | Sear Steak | Mean |
|---|---|---|---|---|---|---|---|
| 3DGStream [35] | 26.73 ±0.05 | 31.38 ±0.04 | 31.36 ±0.15 | 27.45 ±0.01 | 31.56 ±0.50 | 32.44 ±0.02 | 30.15 ±0.13 |
| **HiCoM** (ours) | 28.04 ±0.29 | 32.45 ±0.13 | 32.72 ±0.25 | 28.37 ±0.18 | 32.87 ±0.12 | 32.57 ±0.52 | 31.17 ±0.25 |

Table 9: **Per-scene quantitative results on the PanopticSports dataset.** The asterisk (*) indicates that our method employed the same color trick as Dynamic 3DGS.

| Method | Juggle | Boxes | Softball | Tennis | Football | Basketball | Mean |
|---|---|---|---|---|---|---|---|
| Dynamic 3DGS [25] | 29.48 | 29.46 | 28.43 | 28.11 | 28.49 | 28.22 | 28.70 |
| 3DGStream [35] | 24.68 | 23.69 | 22.44 | 23.09 | 24.97 | 20.01 | 23.15 |
| **HiCoM** (ours) | 27.82 | 27.37 | 28.06 | 27.80 | 27.80 | 27.29 | 27.69 |
| **HiCoM**$^\star$ (ours) | 29.55 | 28.96 | 29.08 | 29.02 | 29.25 | 28.60 | 29.08 |

## B.2 Per-Scene Quantitative Results

We present the quantitative results for each scene from two datasets respectively in Tables 6 and 7 to provide a more detailed comparison. We also provide the results of several state-of-the-art offline methods on N3DV dataset in Table 6 as a reference. Among these, Coffee Martini and Flame Salmon are particularly complex scenes that require a greater number of 3D Gaussians and motion parameters in our HiCoM framework, resulting in relatively higher storage requirements. However, they still require significantly less storage than training with the 3DGStream method.

Table 10: **Quantitative results on the Particle-NeRF dataset.**

| Method | Particle-NeRF [25] | Dynamic3DGS [25] | **HiCoM** (ours) |
|---|---|---|---|
| PSNR (dB ↑) | 27.47 | 39.49 | 31.05 |

In addition, we conduct experiments on the PanopticSports dataset, which involves large-scale motions, and compare with the state-of-the-art Dynamic3DGS method, as shown in Table 9. Notably, Dynamic3DGS heavily relies on segmentation masks, while our HiCoM does not require such additional supervision. To ensure a fair comparison, we set the SH to 1. Additionally, we observe that Dynamic3DGS employs a trick to mitigate potential multi-view color inconsistencies by learning a scale and offset parameter for each color channel of each camera in the training views. To enhance performance, we incorporated this same color trick into HiCoM. The results demonstrate that our method significantly outperforms 3DGStream in terms of PSNR. While slightly behind Dynamic3DGS initially, with the same color trick applied, HiCoM clearly surpasses Dynamic3DGS, highlighting its superiority in handling large-scale motion scenarios.

All our experiments were conducted three times using the same random seed. In Table 8, we present the standard deviation of the three runs on the N3DV dataset, showing the fluctuation in PSNR metric, which is relatively minor. We also tried using different random seeds, and the performance fluctuation was still within the range shown in Table 8.

Furthermore, we extended our evaluation to the Particle-NeRF [47] dataset, which features synthetic scenarios with uniform and periodic motions. The Table 10 shows that our HiCoM outperforms Particle-NeRF, demonstrating its ability to effectively manage these types of motion.

## B.3 Additional Qualitative Results

We provide additional qualitative results for other scenes in the N3DV dataset, three scenes from the Meet Room dataset and six scenes of PanopticSports dataset, as shown in Figures 5, 6 and 7, respectively. These results demonstrate the robustness and versatility of our HiCoM framework in capturing and representing dynamic scenes.

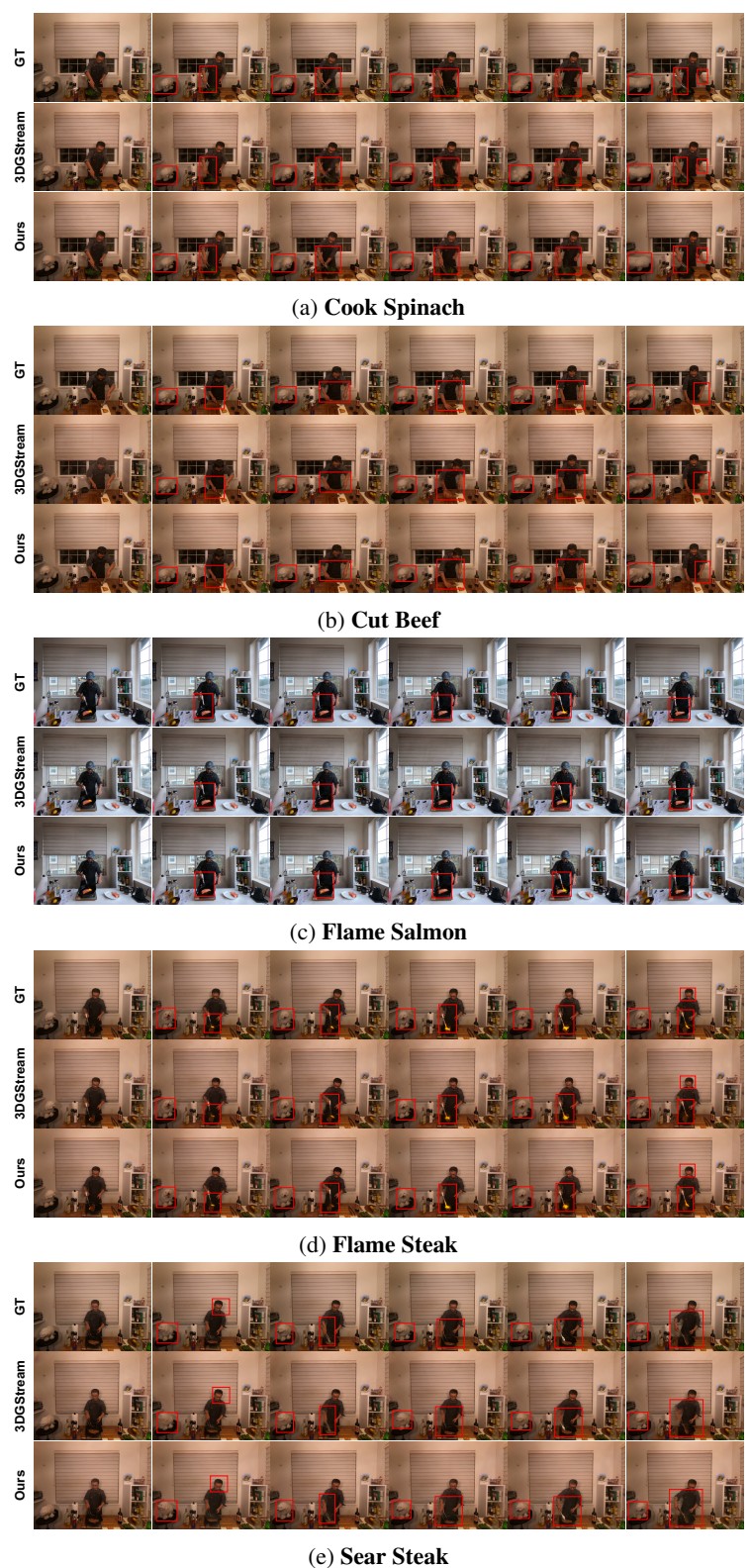

(a) **Cook Spinach**

(b) **Cut Beef**

(c) **Flame Salmon**

(d) **Flame Steak**

(e) **Sear Steak**

Figure 5: **Qualitative results of additional scenes from N3DV dataset.** Frames shown are the $1^{st}$, $61^{st}$, $121^{st}$, $181^{st}$, $241^{st}$, and $300^{th}$ from the test video. Red boxes highlight areas with significant temporal motions.

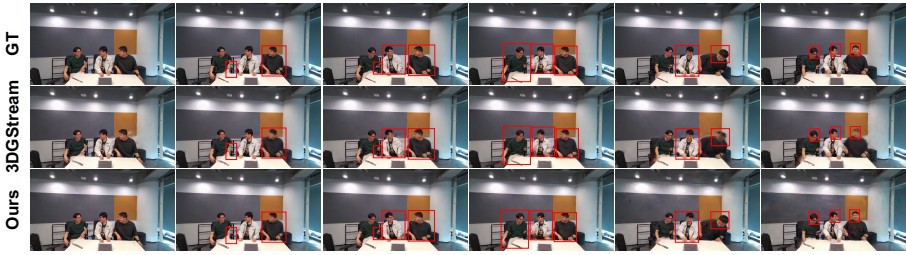

(a) **Discussion**

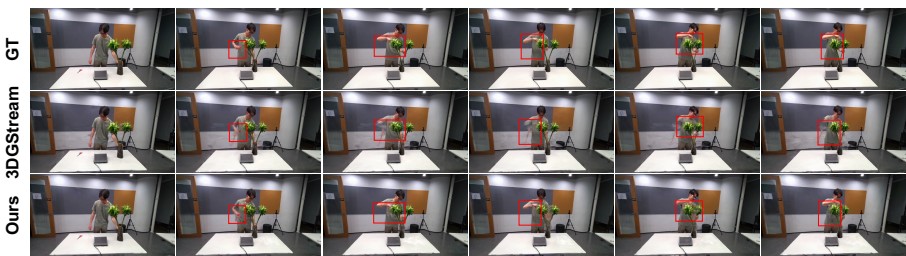

(b) **Trimming**

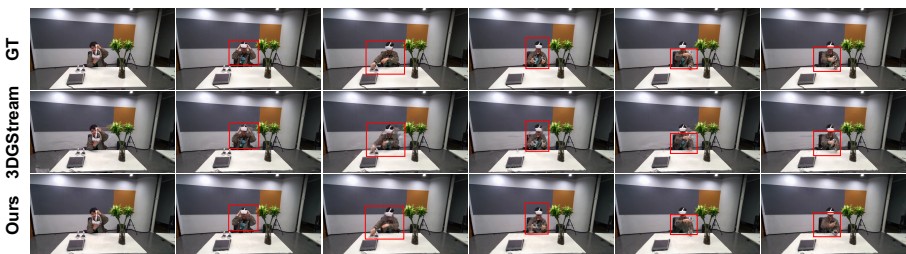

(c) **VR Headset**

Figure 6: **Qualitative results of scenes from Meet Room dataset.** Frames shown are the $1^{st}$, $61^{st}$, $121^{st}$, $181^{st}$, $241^{st}$, and $300^{th}$ from the test video. Red boxes highlight areas with significant temporal motions.

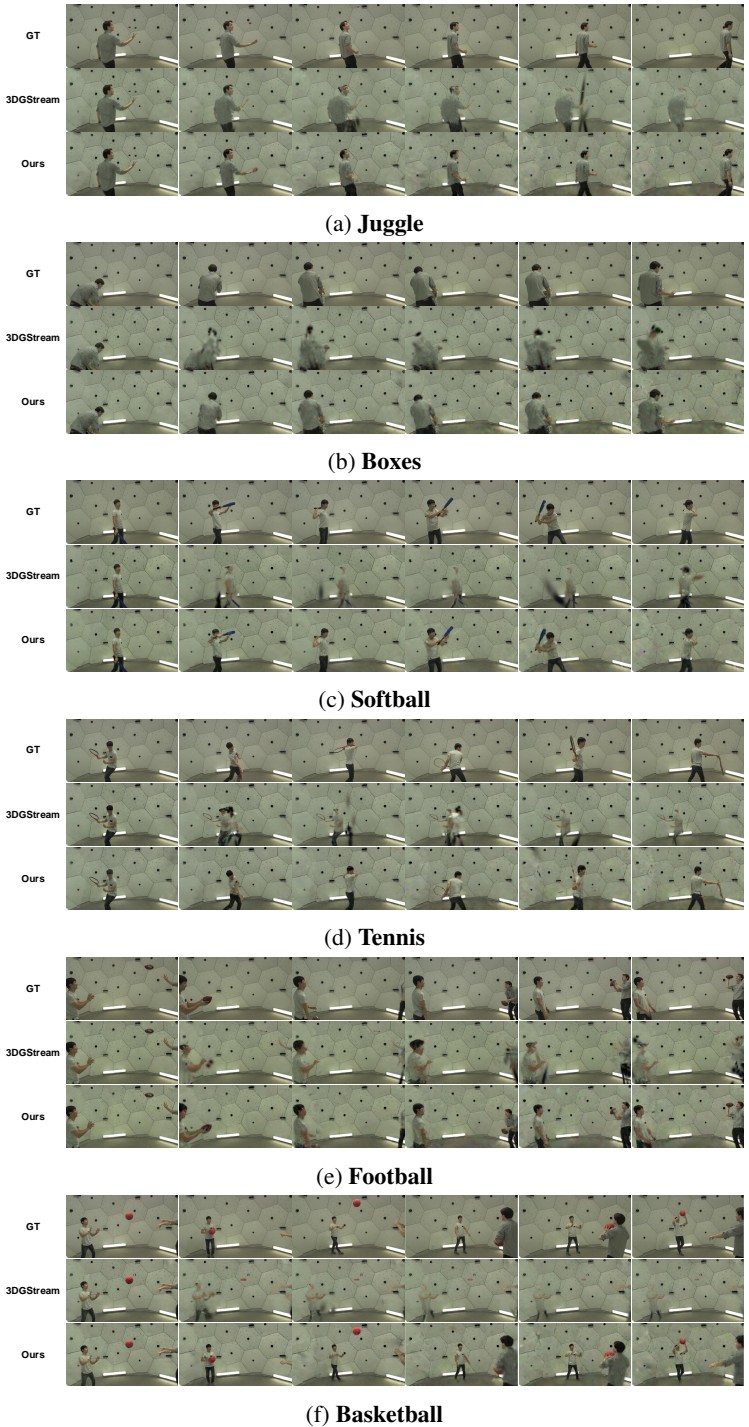

(a) **Juggle**

(b) **Boxes**

(c) **Softball**

(d) **Tennis**

(e) **Football**

(f) **Basketball**

Figure 7: **Visualization of PanopticSports dataset.** Shown are frames 1, 31, 61, 91, 121, and 150 from the $0^{th}$ test video.

