# OpenReview forum: "HiCoM: Hierarchical Coherent Motion for Dynamic Streamable Scenes with 3D Gaussian Splatting"
_NeurIPS.cc/2024/Conference — NeurIPS 2024 poster_

### Official Review · Reviewer_q4mz · 2024-07-03

**Soundness:** 3
**Presentation:** 3
**Contribution:** 3
**Rating:** 6
**Confidence:** 5

**Summary:**

This paper proposes a novel online training framework for multi-view dynamic scenes. A HiCoM framework is introduced for online learning of dynamic scenes. To initialize a better 3D Gaussian, the authors propose adding noise because it may overfit in the forward-facing multiview setups. Modeling motion by a sparse and explicit structure looks very good and also preserves low storage consumption. Experiments also demonstrate that the results are better than previous methods. e.g. 3DGStream.

**Strengths:**

1. A trick about adding noise to 3D Gaussians seems to contribute to forward-facing multiview setups.
2. A hierarchical motion framework is introduced to model the motion in the multiview setups, which seems to work well.
3. Managing deformation by a global and simple data structure is proven to save memory.

**Weaknesses:**

1. It seems that hierarchical coherent motion shows the potential to model large motions. So experiments in the PanopticSports dataset (used in [25]) should be included.
2. Comparison/discussions with Particle-NeRF(Abou-Chakra et al, WACV24) should be included.
3. You can add more comparison figures about ablation studies to demonstrate the results. The reviewer recommends authors submit a video to support a visualization, which is very important in 3D Vision.
4. The introduction section is too long. The reviewer recommends the authors try to express themselves easily and efficiently. Only 3-4 paragraphs (70 lines) are enough to introduce your work. The writing could be improved.

**Questions:**

1. How can you integrate the 3D Gaussians into the initial 3DGS? If you add 3D Gaussians at t(t>0) and how to compute its position at t=0.
2. The reviewer expects to see the results on the Panopticsports dataset. If your code is based on 4D-GS, you can run the experiments directly.

Overall, there are some written problems in the paper and there are lack of visualizations and discussions. The main concern of the reviewer is whether the HiCoM can model large motions such as Panopticsports. If the author can provide positive results on Panopticsports and more discussions, the reviewer may recommend accepting this paper.

**Limitations:**

The limitations are shown in the paper.

---

> ### Author Rebuttal · Authors · 2024-08-06
>
> Thank you for positive assessment of adding noise to 3D Gaussian representation learning, the effectiveness of our hierarchical coherent motion, and the simple deformation data structure to save memory. We now address the main concerns you have raised as follows.
>
> ### 1. The performance of HiCoM in reconstructing scenes with large motions.
>
> As per your suggestion, we conducted experiments on the PanopticSports dataset. We set SH to 1 for fair comparison, and present results in the following table. Our HiCoM significantly surpasses the competitor 3DGStream in terms of PSNR, but slightly lower than Dynamic3DGS [3]. This is primarily because Dynamic3DGS utilizes segmentation masks as additional supervision, substantially boosts performance. According to Dynamic3DGS paper, removing segmentation masks leads to a significant performance drop (over 5dB), resulting in lower performance compared to our method.
> In the attached PDF, we present rendering results of all six scenes from one of the test viewpoints. The 3DGStream struggles with large movements; people and objects in the scene generally remain near their initial positions and gradually become invisible. We believe this is due to the limitations of its grid-based hash encoding and MLP, which cannot effectively handle high-frequency temporal motion information. In contrast, our HiCoM captures large movements of people and objects in the scene well, demonstrates sufficient flexibility, but there is still considerable room for improvement in reconstruction quality, especially for smaller moving objects.
>
>
> | Method          | Juggle | Boxes | Softball | Tennis | Football | Basketball | Mean   |
> |-----------|--------|-------|----------|--------|----------|------------|--------|
> | Dyanmic3DGS      | 29.48  | 29.46 | 28.43    | 28.11  | 28.49    | 28.22      | 28.70  |
> | Dyanmic3DGS (w/o mask) | 24.14  | -     | -        | -      | -        | -          | -      |
> | 3DGStream          | 24.68  | 23.69 | 22.44    | 23.09  | 24.97    | 20.01      | 23.15  |
> | **HiCoM (ours)**     | 27.82  | 27.37 | 28.06    | 27.80  | 27.80    | 27.29      | 27.69  |
>
>
> ### 2. Comparison with Particle-NeRF.
>
> After carefully reviewing Particle-NeRF paper, we found it to be an excellent piece of work. It uses discrete particles for scene representation instead of the uniform grids used in previous NeRF work (InstantNGP). By optimizing particle positions and features through gradients, it achieves a similar effect to 3DGS, with the main difference being in the rendering pipeline. Particle-NeRF models motion through particle movement, focusing more on periodic, regular, and micro-level object motions. In contrast, our method targets general, scene-level dynamic changes.  As we know, some Gaussian Splatting works  [1, 2] also specialize in periodic and regular object motions. We believe Particle-NeRF might not be as effective in modeling the dynamic changes that our method addresses.
>
> Despite this, we conducted experiments on the Particle-NeRF dataset, including five scenes, each with 40 viewpoints. We used 30 for training and the remaining 10 for testing. Please note that the Particle-NeRF paper mentions using only 20 training viewpoints and 10 testing viewpoints. As shown in the table below, our method achieves good reconstruction quality. Unfortunately, up to the time of submitting this response, we were unable to successfully run their open-source code, preventing us from conducting a more detailed comparison on the real-world dynamic scenes used in our paper.
>
> | Method  | PSNR  |
> |----|---|
> | Particle-NeRF       | 27.47 |
> | Dynamic3DGS         | 39.49 |
> | **HiCom (ours)**    | 31.05 |
>
> ### 3. Add more comparison figures and visualization videos.
>
> Due to space limitations in the main manuscript, we have provided visual comparisons of all other scenes with the 3DGStream method in the supplementary materials. We have created videos to support our visualizations and are very willing to provide them. However, due to the review guidelines, we are unable to include external links here. We will consider including the comparison figures from the ablation studies in the revised manuscript and attach relevant visualization videos.
>
> ### 4. The introduction section is too long.
>
> We appreciate the reviewer's feedback on the length and clarity of the introduction section. Our introduction in the manuscript spans from line 23 to line 95, totaling 72 lines, which is in line with the 70 lines recommended by the reviewer. Nevertheless, we will further refine this section in the final version to ensure it is as concise and precise as possible.
>
> ### 5. Integrate the 3D Gaussians into the initial 3DGS.
>
> In our online reconstruction framework, we add some new Gaussian primitives to benefit the reconstruction and rendering of the current frame, but does not require computing their positions at $t=0$. Our primary objective is dynamic scene reconstruction and rendering, rather than object tracking.
>
> We sincerely appreciate your detailed and constructive feedback. Your insights have been invaluable in refining our work, and we are committed to addressing your concerns to improve the overall quality of our paper. Thank you once again for your time and effort in reviewing our manuscript. We look forward to any further comments you may have.
>
> [1] Yutao Feng,Xiang Feng,Yintong Shang, Ying Jiang, Chang Yu, Zeshun Zong,Tianjia Shao,Hongzhi Wu, Kun Zhou, Chenfanfu Jiang, et al. Gaussian splashing: Dynamic fluid synthesis with gaussian splatting. arXiv preprint arXiv:2401.15318, 2024.
>
> [2] Licheng Zhong, Hong-Xing Yu, Jiajun Wu, and Yunzhu Li. Reconstruction and simulation of elastic objects with spring-mass 3d gaussians. In Proceedings of the European Conference on Computer Vision (ECCV), 2024.
>
> [3] Jonathon Luiten, Georgios Kopanas, Bastian Leibe, and Deva Ramanan. Dynamic 3d gaussians: Tracking by persistent dynamic view synthesis. In International Conference on 3D Vision (3DV), 2024.

---

> > ### Comment · Reviewer_q4mz · 2024-08-08
> >
> > Thanks for your comprehensive answers, most of my problems are solved. However, I still have two questions:
> >
> > 1. Can HiCoM be optimized with masks? In dynamic scene reconstruction, filtering dynamic objects with a foreground mask could be a normal and correct solution.
> >
> > 2.  ”comparison with particle-NeRF" is strange because I think rigid & regular motion is the subset of scene level and non-rigid motion.
> >
> > Therefore, I would like to expect the author's and other reviewer's comments on this issue.

---

> > > ### Author Response · Authors · 2024-08-09
> > >
> > > Thank you for your comment. We will now address the two questions you raised.
> > >
> > > ### 1. Can HiCoM be optimized with masks?
> > >
> > > Our HiCoM is indeed extendable and can be optimized with masks. However, our experiments have shown that this does not yield additional performance gains. Specifically, we incorporated the loss between the rendered mask and the ground truth mask (referred to as $L_{mask}$), and constrained the background Gaussian primitives to remain as static as possible (referred to as $L_{static}$), as suggested by Dynamic3DGS.
> > >
> > > Despite these efforts, our method has already shown an ability to learn the motion of the foreground and background reasonably well, as demonstrated in the attached PDF. The masks are rendered with sufficient accuracy, and further optimization using masks did not result in significant improvements. Moreover, since our method inherently learns minimal background motion, the $L_{mask}$ and $L_{static}$ made limited contributions.
> > >
> > > ### 2. "comparison with particle-NeRF" is strange.
> > >
> > > We are unclear if the reviewer's concern is with the experimental results or our conclusion about ParticleNeRF. We will clarify both aspects briefly.
> > >
> > > Firstly, regarding the experimental results, the ParticleNeRF dataset comprises synthetic objects with relatively simple but precise and regular motions (e.g., wheel rotating) and without changes in appearance.
> > > Our HiCoM potentially performs better than ParticleNeRF due to several factors, including the inherent advantages of Gaussian Splatting, the general compatibility of our motion learning mechanism with this dataset, and the additional Gaussian primitives introduced at each frame. Dynamic3DGS learns individual positions and rotations of Gaussian primitives at each time step, providing finer detail compared to our approach, where multiple Gaussian primitives share the same motion. Therefore, it is reasonable that Dynamic3DGS outperforms our HiCoM in this context.
> > >
> > > Secondly, our conclusions about ParticleNeRF and its motion modeling are based on its paper and experiments. ParticleNeRF's position-based dynamics (PBD) algorithm tends to distribute particles more uniformly (but unlike the fixed grid of InstantNGP), while effective for simpler synthetic datasets with periodic, regular and micro motions, may struggle to handle the intricate details required in real-world dynamic scenes, where complex regions often need denser particles or Gaussian primitives and more detailed motion. ParticleNeRF's e experimental results demonstrate that it performs well on its own synthetic ParticleNeRF and Animated Blender datasets, both involving object-level motions, but the Figure S6 in its supplementary material shows less satisfactory results on a full scene. These observations support our conclusion.
> > >
> > > Thank you again for your valuable feedback. We hope our responses meet your expectations. If there are any further questions or concerns, please do not hesitate to let us know.

---

> ### Comment · Reviewer_q4mz · 2024-08-13
>
> Dear authors, thanks for your reply. My concerns still exist.
>
> 1. As the authors addressed, if HiCoM can optimized with masks why it doesn't improve the final PSNR?  What's the reason? I don't think it would demonstrate HiCoM can achieve better foreground rendering quality compared with Dynamic 3DGS.
>
> 2. As the author said, Particle-NeRF shows its advantages in uniform motions, but I think it is the subset of real-world dynamics. I think a dynamic representation can handle the complex motions in real-world scenes, it can also handle uniform and periodic motion. Dynamic 3D Gaussians also demonstrate that it would achieve good results in both particleNeRF dataset and its used Sports dataset.
>
> Therefore, I think HiCoM should beat dynamic3DGS and particleNeRF in both synthetic animation datasets and panopticSports datasets, if evaluation metrics cannot demonstrate that, a video or figure is also convincible.

---

> ### Author Response · Authors · 2024-08-13
>
> Dear Reviewer,
>
> Thank you for your comments. We would like to acknowledge that our method  does indeed perform slightly worse than Dynamic3DGS on the two datasets you mentioned, where the appearance shows less variation and no dynamic addition or removal of content compared to the initial frame.
>
> Beyond using masks, Dynamic3DGS includes the motion parameters for **each Gaussian primitive** at every time step, whereas our method allows **Gaussians within a local region** to share same motion parameters. While this may lead to a slight compromise in detail compared to Dynamic3DGS, it offers significant advantages in storage and training efficiency for online dynamic scene reconstruction.
>
> We would like to further address your specific concerns as follows.
>
> ### 1. Why optimizing with masks doesn't improve the final PSNR of our HiCoM?
>
> Masks represent the relative position of the foreground **as a whole** to the background, and they can be helpful in learning **the overall motion of foreground**. Our method already learns the overall foreground motion well without masks, demonstrated by the consistency between the rendered and the ground truth masks (will be included in the revised manuscript ). Therefore, using masks does not enhance the overall foreground motion learning. In contrast, Dynamic3DGS struggles with overall foreground motion without masks, highlights our method's strength in capturing large global motion.
>
> While we effectively position the whole foreground, Dynamic3DGS’s ability to optimize each Gaussian’s position at different time steps leads to better internal detail and higher PSNR.
> However, we aim for our online dynamic reconstruction method to still perform well without masks, as in many cases, masks may not always be readily available.  We believe that applying finer division to the foreground region in our framework could further improve the quality.  While this is beyond the scope of the current work, it remains a promising direction for future research.
>
> ### 2. Performance comparison on the ParticleNeRF dataset.
>
> The experimental results we presented show that our method performs better than ParticleNeRF, indicating that our approach is **indeed capable** of handling the uniform and periodic motions present in this synthetic dataset. The reviewer's expectation that our method should also surpass Dynamic3DGS may stem from a misunderstanding. As we previously explained, Dynamic3DGS benefits from its ability to optimize each Gaussian's position, which gives it an advantage, especially in these simpler synthetic scenarios.
>
> We hope this clarification helps to address your concerns, and we sincerely appreciate your valuable feedback. We will make sure to thoroughly incorporate these clarifications into the revised version of our paper. Unfortunately, according to the **review guidelines**, we are unable to provide external links at this stage, but we would be happy to present the video results to you in a compliant manner at an appropriate time.
>
> Thank you once again for your consideration.
>
> Best regards,
>
> The Authors

---

> > ### Comment · Reviewer_q4mz · 2024-08-13
> >
> > Thanks for your valuable comments. These are my consideration.
> >
> > 1. I do not doubt that HiCoM can beat 3DGStream in most evaluation metrics and cases (positive).
> >
> > 2. Currently, Dynamic 3DGS is indeed better at solving large motions with foreground masks. And I think optimizing dynamic scenes with masks is a necessary step because it's easy and cheap to get a foreground mask. But HiCoM seems to be an online optimizing method, that enjoys less training times and less storage costs. If the author can demonstrate that HiCoM can beat dynamic 3DGS from any perspective I would raise my score. The current rebuttal can only answer my questions. (borderline)
> >
> > 3. Hope the author can add all the results in both the synthetic animation dataset and PanopticSports dataset in the revised version, both are important benchmarks in evaluating a method.
> >
> > 4. If possible, release the code.
> >
> > Therefore, I would like to keep my score.

---

> > > ### Author Response · Authors · 2024-08-13
> > >
> > > Dear Reviewer,
> > >
> > > Thank you for your thoughtful consideration.
> > >
> > > We acknowledge that the first point has been addressed to your satisfaction, and we will certainly include the results on both the synthetic animation dataset and the PanopticSports dataset as you requested in the third point. Regarding the forth point, we are always committed to releasing our code to benefit the community and advance research in this field.
> > >
> > > Now, focusing on your second point. After further investigation, we identified **a key trick** that contributed to Dynamic3DGS's superior performance. Specifically, Dynamic3DGS learns **a scale and offset parameter** for each color channel for each camera in the training views. Although this trick was mentioned in its paper, it was not emphasized or subjected to ablation studies.
> > >
> > > Initially, we overlooked this trick. However, after implementing this trick in our HiCoM, we observed **significant improvements**. For example, in the Football scene, our HiCoM achieved an average PSNR of **29.25 dB** over all 150 frames, which exceeds Dynamic3DGS's **28.49 dB**. Moreover, when this trick was removed from Dynamic3DGS, the PSNR dropped by **4 dB**. This further demonstrates that the reconstruction quality of the initial frame in online learning has a significant impact on overall performance. Our latest experimental results also validate our earlier response to you, indicating that the role of masks is more beneficial for learning overall foreground motion rather than being the key to achieving better foreground rendering quality. Of course, this aspect requires further in-depth study in future research. The table below presents more comprehensive experimental results. With the trick, our **HiCoM significantly outperforms Dynamic3DGS** across five out of the six scenes. We will further analyze these results and include in the revised paper.
> > >
> > > | Method                 | Juggle | Boxes | Softball | Tennis | Football | Basketball | Mean  |
> > > |------------------------|--------|-------|----------|--------|----------|------------|-------|
> > > | Dynamic3DGS            | 29.48  | **29.46** | 28.43    | 28.11  | 28.49    | 28.22      | 28.70 |
> > > | 3DGStream              | 24.68  | 23.69 | 22.44    | 23.09  | 24.97    | 20.01      | 23.15 |
> > > | **HiCoM (ours)**       | 27.82  | 27.37 | 28.06    | 27.80  | 27.80    | 27.29      | 27.69 |
> > > | **HiCoM (w/ trick)**   | **29.55**  | 28.96 | **29.08**    | **29.02**  | **29.25**    | **28.60**      | **29.08** |
> > >
> > >
> > > We hope this new evidence addresses your concerns. We believe that your valuable feedback has significantly improved the **solidity** of our work, and we are very grateful for that. If you have no further concerns, we hope you will consider raising your score.
> > >
> > > Best regards,
> > >
> > > The Authors

---

> > > > ### Comment · Reviewer_q4mz · 2024-08-14
> > > >
> > > > Thank you for your solid reply. I have no more questions regarding HiCoM.
> > > >
> > > > I wish the HiCoM could be a new baseline for evaluating multiview & streamable dynamic scene NVS methods.
> > > >
> > > > Finally, I expect to see the code will be released if the paper is accepted for reproducing all experimental results. If the author promise to do that, I would like to change my score.

---

> > > > > ### Author Response · Authors · 2024-08-14
> > > > >
> > > > > Dear Reviewer,
> > > > >
> > > > > Thank you for your encouraging words. We also deeply appreciate the efforts and contributions you and the other reviewers have made to improve our work.
> > > > >
> > > > > We assure you that we will release the code upon acceptance of the paper. We have put significant effort into ensuring that our code is clean, easy to maintain, and extensible. We hope that once it’s open-sourced, it will be a useful resource for the research community and facilitate fair comparisons across different methods.
> > > > >
> > > > > Thank you again for your support.
> > > > >
> > > > > Best regards,
> > > > >
> > > > > The Authors

---

> > > > > > ### Comment · Reviewer_q4mz · 2024-08-14
> > > > > >
> > > > > > Many thanks.
> > > > > >
> > > > > > I will change my score to `weak accept`.
> > > > > >
> > > > > > Best wishes.

---

> > > > > > > ### Author Response · Authors · 2024-08-14
> > > > > > >
> > > > > > > We appreciate your time and effort, as well as your final recognition of our improved work.
> > > > > > >
> > > > > > > Sincerely.

---

### Official Review · Reviewer_QKxo · 2024-07-06

**Soundness:** 3
**Presentation:** 3
**Contribution:** 2
**Rating:** 5
**Confidence:** 4

**Summary:**

This paper proposes a novel framework for online reconstruction of dynamic scenes based on 3DGS. The main contribution lies in its Hierarchical Coherent Motion, which gives a more compact motion representation. Results show its effectiveness and rendering improvement.

**Strengths:**

1. The method is straightforward and easy to follow.

2. The pertubation smoothing strategy is simple but effective. I think it maybe also work in sparse view rendering of 3DGS.

3. The ablation is adequate.

**Weaknesses:**

1. Continual Refinement seems similar to the strategy in 3DGStream. It would be better to clarify the difference.

2. Most parts of the scene is static, uniformally dividing the scenes to model the motion seems redudant.

3. Embedding the motion in space alone may have limited modelling ability, especially when an object undergoes complex movements within a small region, such as intricate transformations back and forth. It might be better to incorporate the temporal dimension.

4. Although the method is tailored for streamable videos, it would be better to compare with other dynamic 3DGS methods, such as 4DGaussians, GaussianFlow, spacetime Gaussian. The previous work 3DGStream also includes the comparison with offline or non-streamable methods.

**Questions:**

1. Since the work focus on the online reconstruction of streamable videos, the efficiency is very important. I am wondering whether the cost of initialization from SfM (Colmap) should be considered.

**Limitations:**

1. No ethic issues.

2. The authors kindly point out their limitations in the paper. The initialization of 3DGS in the first frame is significant and the paper is not fully addressed. It would be kind for authors to show how the initial frame learning influence the whole pipeline currently, as they claim it is not fully addressed.

---

> ### Author Rebuttal · Authors · 2024-08-06
>
> We greatly appreciate your positive feedback on the clarity and straightforward nature of our method. We are particularly encouraged by your recognition of the effectiveness of our perturbation smoothing strategy, as well as your appreciation of the simplicity. Your acknowledgment of the thoroughness of our ablation study is also highly valued. We have carefully reviewed your questions and concerns and address them in detail below.
>
> ### 1. The difference between our continual refinement and 3DGStream.
>
> Our continual refinement may appear similar to the second stage of 3DGStream in form, but it fundamentally differs in approach. 3DGStream continuously applies learned motion to the Gaussian representation derived from the first frame, with newly added Gaussian primitives only used for rendering the current frame and not passed to subsequent frames. In contrast, our method merges new Gaussian primitives into the initial Gaussian representation, then removes the same number of less important Gaussian primitives based on importance metrics such as the opacity we used in this work or other indicators. If an object does not exist in the first frame, 3DGStream adds Gaussian primitives in every subsequent frame to fit the object, while our method can pass these Gaussian primitives to the next frame, eliminating the need to refit the object.
> We believe our design is superior and will clearly state this difference in the revised manuscript to reduce any potential confusion for readers.
>
> ### 2. The uniformly dividing the scene seems redudant.
>
> Thank you for highlighting this issue. However, our method demonstrates that uniformly dividing the scene is simple and effective. We agree that adaptively dividing the scene based on the granularity of motion could further improve the motion representation: regions with minimal motion could be coarsely divided, while areas with complex motion could be finely partitioned. Based on your suggestion, we will explore more effective scene division methods in future work.
>
> ### 3. Incorporate the temporal dimension into motion.
>
> Our online reconstruction framework focuses on representing and learning changes between adjacent frames in general real-world scenes, which keeps our method relatively simple but has proven effective in experiments. We agree that embedding the motion in space alone may have limited modeling ability, especially for complex movements such as intricate transformations within a small region. We acknowledge that some existing works specifically address such periodic motion patterns [1, 2]. Future work could enhance our framework by incorporating temporal information from observed frames to construct more comprehensive motion patterns.
>
>
> ### 4. Comparison with offline reconstruction methods.
>
> We included data reported by state-of-the-art offline dynamic scene reconstruction methods in the supplementary materials. Specifically, Table 8 in the Appendix shows that our method is comparable in reconstruction quality  across nine scenes from two datasets. We will highlight these comparisons more prominently and incorporate additional recent works in our revised manuscript.
>
> ### 5. The cost of initialization from SfM.
>
> The cost associated with SfM does not significantly affect the online reconstruction workflow. In dynamic scene reconstruction, the viewpoints are typically sparse, meaning that SfM needs to be applied to only a limited number of views from the initial frame. Generating the initial point cloud usually takes only a few minutes. In practical scenarios, camera positions for video capture are often pre-arranged, allowing the majority of the SfM work to be completed before the reconstruction process begins.
>
> ### 6. The influence of the initial frame learning.
>
> Our online framework reconstructs the scene’s fundamental geometry and appearance through initial frame learning, and assumes that most objects in the dynamic scene will not undergo significant appearance changes. Subsequent learning focuses on representing and learning the motion of objects from the initial frame, adding a few new Gaussian primitives to handle previously unseen content and to improve motion learning. In contrast, if the initial scene's geometry and appearance are poorly reconstructed, it increases the burden on subsequent learning, leading to slower convergence and degraded reconstruction quality, which can accumulate over time.
>
> We are grateful for your constructive feedback, which has been instrumental in enhancing the quality of our manuscript. If our clarifications have addressed your concerns, we kindly ask you to reconsider your evaluation of our manuscript. Should you have any further questions or require additional clarifications, we are happy to provide further details. Thank you once again for your time and consideration.
>
> [1] YutaoFeng,XiangFeng,YintongShang,YingJiang,ChangYu,ZeshunZong,TianjiaShao,HongzhiWu, Kun Zhou, Chenfanfu Jiang, et al. Gaussian splashing: Dynamic fluid synthesis with gaussian splatting. arXiv preprint arXiv:2401.15318, 2024.
>
> [2] Licheng Zhong, Hong-Xing Yu, Jiajun Wu, and Yunzhu Li. Reconstruction and simulation of elastic objects with spring-mass 3d gaussians. In Proceedings of the European Conference on Computer Vision (ECCV), 2024.

---

> ### Author Response · Authors · 2024-08-12
>
> Dear Reviewer QKxo,
>
> I hope you’re doing well.
>
> We understand you may have a busy schedule and might not have had the chance to review our rebuttal yet. We hope that our responses have adequately addressed your concerns. If you have any further questions or need additional information, please feel free to let us know.
>
> Thank you again for your time and effort in reviewing our submission.
>
> Best regards,
>
> The Authors of Submission 809

---

> ### Comment · Reviewer_QKxo · 2024-08-13
>
> Thanks for authors' reply and effort. I have read the rebuttal and discussion with other reviewers. Most of my concerns are addressed. However, for the comparison with offline reconstruction methods, I hope the author can clearly present the comparison of methods in a table. Table 8 of the supplementary materials mentioned by the author only shows the results of offline methods, which made me spend more effort to compare the results of this method again. It is acceptable that offline methods perform better than online methods because they invest more time and even storage to optimize.
>
> Besides, I have one more concern. I found the reported result for 3DGStream is much lower than its original paper, especially for the meet room dataset (For N3DV, the 3DGStream reports **31.67** PSNR while the authors' reimplemented verision reports **30.15** PSNR. For meet room, the 3DGStream reports **30.79** PSNR while the author's reimplemented verision reports **25.96** PSNR). I really appreciate authors' effort for reimplementation and I understand that the open-source code may not have been systematically maintained. However, such performance differences might make it difficult to conduct a fair comparison in the future works, as two works placed different results on the same dataset. Has the author tried contacting the authors of 3DGStream? I am not sure if more discussion is needed at this point. It is just my concern.

---

> ### Author Response · Authors · 2024-08-13
>
> Dear Reviewer,
>
> Thank you for your detailed comments. We greatly appreciate your insights.
>
> We agree with your suggestion and will follow your advice to present the results of all methods in a single table, making it easier for readers to compare and analyze.
>
> Regarding the discrepancy in the results of 3DGStream, we consulted with the 3DGStream authors before they released the code. They confirmed that they did not use the original camera poses provided by the dataset and applied distortion correction to the views. During this work, we followed the data processing practices used by 4D-GS (CVPR 2024, in Table 8) and several previous works using this dataset, including the StreamRF (the paper that originally proposed the Meet Room dataset) in our Table 1, all of which utilize the original camera poses. Additionally, we noted in the Appendix that the processed views in 3DGStream contain approximately 50,000 fewer pixels than the original views. However, from our observations of the N3DV dataset, we did not notice significant distortion. Therefore, to ensure a fair comparison, we chose to remain consistent with the practices used by previous methods, using the original view images.
>
> After submitting our paper, we also conducted experiments using the undistorted dataset following the open-source code provided by the 3DGStream authors. On the Meet Room dataset, the average PSNR was **28.32 dB**, which is significantly lower than the **30.79 dB** reported in their paper. Meanwhile, our HiCoM achieved a PSNR of **31.02 dB** on the undistorted dataset. Following your suggestion, we will include these results in the paper to provide a more comprehensive comparison.
>
> Thank you once again for your thoughtful feedback. If you have any further questions or concerns, please feel free to reach out, and we will do our best to address them. We will also carefully incorporate the discussions with you and other reviewers into the revised manuscript.
>
> Best regards,
>
> The Authors

---

> > ### Comment · Reviewer_QKxo · 2024-08-13
> >
> > Thanks for authors detailed explanation. Hope the authors can add all the results and discussions in the paper or the appendix, which really promotes the development of the community. According to the current rebuttal and other reviews, I would like to keep my score.

---

> > > ### Author Response · Authors · 2024-08-14
> > >
> > > Dear Reviewer,
> > >
> > > Thank you very much for your feedback and for acknowledging our detailed explanations. We appreciate your recognition of the importance of including all results and discussions to benefit the research community, and we will **certainly do so** in the paper or appendix.
> > >
> > > Thanks to the constructive comments from all reviewers, we have worked hard to improve our work. Particularly, driven by Reviewer q4mz’s insightful questions, we conducted additional experiments that demonstrated our method’s superior performance over the state-of-the-art Dynamic3DGS on the PanopticSports dataset with large motions. **Our experimental results and solid analysis has been validated by Reviewer q4mz, who has already raised his/her score based on our new results.**
> > >
> > > We kindly suggest that you refer to our discussions with Reviewer  q4mz, or feel free to discuss the matter directly with he/she, to further assess the improvements we have made. If you have no further concerns, we would be grateful if you could consider raising your score as well.
> > >
> > > Thank you again for your time and effort.
> > >
> > > Best regards,
> > >
> > > The Authors

---

> ### Public Comment · ~Jiakai_Sun1 · 2024-11-18
> **Concerns about the results of 3DGStream on the MeetRoom dataset**
>
> Dear Authors/Reviewers,
>
> I am Jiakai Sun, the first author of the 3DGStream paper. I am delighted to see that this paper on OpenReview recognizes our work as the previous state-of-the-art in this field and has made efforts to re-implementation it.
>
> I would like to clarify a point: the results reported in our original paper are specifically based on the "discussion" scene in the MeetRoom dataset. This is consistent with the settings for StreamRF, as confirmed by the authors of StreamRF (https://github.com/AlgoHunt/StreamRF/issues/13). However, the authors of HiCoM appear to have misunderstood this and assumed that our reported results represent the average across all three scenes. This misunderstanding led to the claim in their response:
> > On the MeetRoom dataset, the average PSNR was **28.32** dB, which is significantly lower than the **30.79** dB reported in their paper.".
>
> However, as can be seen in the Table7 in the OpenReview version of HiCoM , the result is **30.06** dB when they use the open-source code provided by the us, which is much closer to the metric we reported in our paper.
>
> Since OpenReview serves as an open platform for paper reviews, I believe it is important to leave this clarification here to address potential confusion for future researchers who may encounter this discussion.
>
> Thank you.

---

> > ### Comment · Area_Chair_nn7c · 2024-11-18
> >
> > Dear Jiakai,
> >
> > Thanks for the classification. To ensure a consistent comparison, we would either need you to provide an average score across all three scenes or have the HiCoM authors report a comparison for the "discussion" scene only.
> >
> > @Jiakai, Do you think it’s possible to provide the average scores?
> >
> > @HiCoM authors, please track this discussion.
> >
> > - Your AC

---

> ### Public Comment · ~Jiakai_Sun1 · 2024-11-19
>
> Dear AC,
>
> Thank you for your prompt response. We have reached an agreement with the authors of HiCoM on how to update the instructions in our open-source repository's README, as well as enhance the descriptions in their preprint, potential revisions, and open-source code, to provide clearer context for future researchers.
>
> Regarding the scores on MeetRoom, after confirming with the authors of StreamRF (prior to the CVPR2024 submission), we only reported the results for the `discussion` scene in our paper to ensure a fair comparison. The authors of HiCoM will also conduct a secondary confirmation with the authors of StreamRF. We believe that the metrics reported by the authors of HiCoM in Table 7 using our open-source code on the undistorted images (`discussion`: 30.06, `trimming`: 28.82, `VRHeadset`: 29.26) are credible and can be adopted by future work. If StreamRF confirms the use of an average across multiple scenes, we will update our draft to provide both the average scores and scores for each individual scene.
>
> My comment here is to clarify and prevent potential misunderstandings about our work arising from **this statement** in OpenReview:
>
> > On the MeetRoom dataset, the average PSNR was 28.32 dB, which is significantly lower than the 30.79 dB reported in their paper.
>
> I believe that our discussion here, along with planned updates to our (i.e., both 3DGStream and HiCoM) open-source repositories and papers will resolve this issue and provide clearer background information for future researchers.
>
> Thank you again for your attention to this matter.
>
> Best regards,
>
> Jiakai Sun

---

### Official Review · Reviewer_oHau · 2024-07-12

**Soundness:** 3
**Presentation:** 3
**Contribution:** 2
**Rating:** 4
**Confidence:** 4

**Summary:**

This paper deals with the online reconstruction of dynamic scenes from multi view video input. The authors build their method on the popular 3D Gaussian Splatting technique. To tackle this setting, they propose 1. A perturbation smoothing that introduces small perturbations to the 3D positions of the Gaussians to improve the initial reconstruction learned on the first timestep, 2. A hierarchical coherent motion strategy that divides 3D Gaussians into a hierarchical voxel grid and assigns motion coefficients (3D translation and 4D Quaternion rotation) to each voxel, where the motion of each 3D Gaussian is the sum of all voxel-level motions it is associated with.

**Strengths:**

- The perturbation smoothing strategy is highly effective given the results in Tab.2 and Fig. 4
- The method shows state-of-the-art performance on both N3DV and Meet Room
- The method is simple and (partially) effective, the paper is well written

**Weaknesses:**

- The motion levels seem not to significantly affect the performance (cf. Tab. 5), which begs the question of whether the HiCoM strategy is critical for performance. Also, scene motion is piecewise constant, but has sharp boundaries. What if both dynamic foreground and static background fall into the same voxel?
- In L274, the authors mention that mean results over 3 runs are used in the experiments. In L697, the authors mention they do not include error bars. What’s the reason for this?
- Finally, I am curious about why online reconstruction is needed if both multi-view video and precomputed input poses are needed?

**Questions:**

It would be great if the authors could clarify my questions raised in the weaknesses section. Additionally, Equation 6 suggests that the noise is drawn from a normal distribution with variance 1, which seems quite large. Is this an error or is the perturbation actually that large? Are the reconstructions in metric scale?

**Limitations:**

The paper discusses the method limitations sufficiently.

---

> ### Author Rebuttal · Authors · 2024-08-06
>
> Thank you for recognizing the strengths of our work, particularly the perturbation smoothing strategy, the exceptional performance on N3DV and Meet Room datasets, and the quality of our writing. In the following, we address each of your comments and questions in detail.
>
> ### 1. The impact of motion levels on performance.
>
> We need to clarify the meaning of **motion levels** in Table 5. Our method uses three motion levels by default (the 3$^{rd}$ row), which we refer to as "fine," "medium," and "coarse", respectively. When motion levels is set to 1 (the 1$^{st}$ row), only the "fine" level is used. When set to 2 (the 2$^{nd}$ row), both "fine" and "medium" levels are applied. As the table shows, using only the "fine" level already achieves good performance. Adding the "medium" and "coarse" levels provides additional performance gains, *e.g.*, the PSNR of Coffee Martini and  Discussion scenes respectively boosts 0.23 dB and 0.59 dB, but the improvements diminish as the number of levels increases. This indicates that our HiCoM is effective and a few motion levels are sufficient to achieve good results. We will include this clarification in the revised manuscript and add experimental results for single-level motion (present in the table below) to provide a more comprehensive view.
>
>
> | Motion Level                | Coffee Martini | Flame Steak | Discussion |
> |-----------------------------|----------------|-------------|------------|
> | coarse                      | 26.67          | 30.28       | 22.89      |
> | medium                      | 27.24          | 32.14       | 24.91      |
> | fine                        | 27.79          | 32.76       | 26.10      |
> | fine + medium               | 27.94          | **32.88**   | 26.61      |
> | fine + medium + coarse      | **28.04**      | 32.87       | **26.69**  |
>
>
> ### 2. The dynamic foreground and static background fall into the same voxel.
>
> Your observation is indeed insightful and highlights an important consideration in complex scenes. However, Gaussian primitives are not points but ellipsoids, representing small regions in space. We observed that Gaussian primitives for moving objects are generally smaller and densely packed, whereas those for background regions are usually larger and sparsely distributed.
> As long as the finest motion level's coverage area is smaller than the radius of background Gaussian primitives, this issue can be substantially mitigated.
> Additionally, after the motion learning, we will add some new Gaussian primitives, which can further help in correcting inadequately reconstructed regions. Future work could investigate methods to distinguish dynamic objects from the static background, such as filtering based on the size of Gaussian primitive.
>
> ### 3. The absence of error bars.
>
> We performed each experiment three times using the default random seed, but result variations were minimal. Given the number of metrics we reported, there was limited space in the tables. Additionally, some of the data were directly referenced from other papers that did not include standard deviations, so we omitted them for consistency.
> Considering your suggestion, we will include all avaliable standard deviations for PSNR metrics (as shown in the table below) in Tables 6 and 7 in our revised manuscript.
>
>
>
> | Method              | Coffee Martini      | Cook Spinach        | Cut Beef           | Flame Salmon       | Flame Steak        | Sear Steak         | Mean              |
> |---------------------|---------------------|---------------------|--------------------|--------------------|--------------------|--------------------|--------------------|
> | 3DGStream           | 26.73~±~0.05       | 31.38~±~0.04       | 31.36~±~0.15       | 27.45~±~0.01       | 31.56~±~0.50       | 32.44~±~0.02       | 30.15~±~0.13         |
> | **HiCoM (ours)**    | 28.04~±~0.29       | 32.45~±~0.13       | 32.72~±~0.25       | 28.37~±~0.18       | 32.87~±~0.12       | 32.57~±~0.52       | 31.17~±~0.25         |
>
>
>
>
>
> ### 4. The necessity of online reconstruction.
>
> Most methods for dynamic scene reconstruction work in an offline manner, but these methods may suffer from low storage efficiency, high resource demands, and insufficient real-time response for long-duration or real-time interactive dynamic scenes. For example, for a dynamic scene lasting 2 hours, methods using neural networks to model scene motion may require very large neural networks, substantial GPU memory, and may have slow convergence rates. The goal of online reconstruction is to mitigate these issues while achieving competitive reconstruction quality. Online reconstruction does not need to wait for all video capture to complete, and the cost for pose estimation is negligible. Real-time high-fidelity dynamic scene reconstruction would be valuable for modern applications such as AR/VR, free-viewpoint video streaming, and online meeting.
>
> ### 5. The perturbation noise present in Equation 6.
>
> We would like to clarify that Equation 6 also includes a coefficient $\lambda_{noise}$, which adjusts the noise intensity. This coefficient allows us to control the perturbation magnitude, ensuring it is appropriate for the specific context of the reconstruction.
>
> Thank you once again for your valuable feedback and insightful comments. We hope that our responses have addressed your concerns and clarified the points raised in your review. If you have any further questions or require additional clarifications, please feel free to leave a comment. We remain committed to providing any further information you may need.

---

> > ### Comment · Reviewer_oHau · 2024-08-12
> >
> > I thank the authors for the response, most of my concerns are addressed. Could the authors elaborate on "using the default random seed"? Ideally, the experiments should be run with different random seeds to make sure the improvements are significant. Do the added error bars indicate a deviation of different runs with the same random seed? If so, I'd be curious as to why the results differ even when choosing the same random seed.

---

> > > ### Author Response · Authors · 2024-08-12
> > >
> > > Dear Reviewer,
> > >
> > > Thank you for your response, and we are pleased to hear that our rebuttal addressed most of your concerns.
> > >
> > > Regarding the issue of random seeds, we have conducted an additional experiment with a different random seed. The results remain within the range we reported. Randomness is pervasive in experiments, and even with the same random seed, variations can occur due to factors such as GPU scheduling and optimization algorithms. In the field of Gaussian Splatting, almost all works use default random seed because neural networks are not extensively involved, the parameters of Gaussian primitives are not influenced by random seeds. For further discussion on this topic, please refer to issue 89 in the official 3DGS code on GitHub. We will further validate our results with different random seeds and emphasize this in the revised manuscript.
> > >
> > > Thank you once again for your comment. We would be grateful if you could consider re-evaluating our work in light of these clarifications.
> > >
> > > Best regards,
> > >
> > > The Authors

---

> ### Comment · Area_Chair_nn7c · 2024-08-09
> **Discussion**
>
> Dear reviewer oHau,
>
> We received diverse reviews for this submission, and you were initially negative. It would be extremely helpful if you could review the rebuttal and participate in the discussion over the next two days. Thank you very much!
>
> - Your AC

---

### Author Rebuttal · Authors · 2024-08-06

We sincerely thank all the reviewers for their thorough evaluation and valuable feedback on our manuscript. We are pleased that all reviewers recognized strengths of our proposed method and find it simple and effective. Reviewers 1 (oHau), 2 (QKxo) highlighted the effectiveness of our perturbation smoothing strategy, while Reviewers 2, 3 (q4mz) appreciated our hierarchical coherent motion framework. Reviewer 1 specifically noted the state-of-the-art performance of our method on N3DV and Meet Room datasets, and Reviewer 3 praised the memory efficiency of our motion modeling approach. We deeply appreciate these positive comments and have provided detailed responses to each reviewer's suggestions and concerns individually. The attached PDF includes visualization results on the PanopticSports dataset, which features large motion scenes, as suggested by Reviewer 3.

---

### Decision · Program_Chairs · 2024-09-25

**Decision:**

Accept (poster)

**Comment:**

All reviewers agree that the submission proposes a simple and effective solution for radiance field reconstruction from multi-view streaming videos. The authors efficiently addressed the reviewers' concerns in the rebuttal and during the intense discussion. AC recommends acceptance. Please carefully address the concerns raised by the reviewers in the camera-ready version. For example, include the new comparison table with dynamic 3DGS after incorporating the image embedding trick, an ablation study on Motion Level, additional comparison figures and visualization videos, and the release of the code. We look forward to seeing these contributions from the open-source community.